# Semi-Supervised Learning with Noisy Proxy Covariates: Generalization Bounds and Distribution Regression

**Kwangho Kim** [1]  **Jisu Kim** [2]

## Abstract

In many modern machine learning pipelines, abundant pretrained representations serve as noisy proxy covariates, while task-specific labels remain scarce. We study semi-supervised regression in this setting, and propose a simple two stage estimator that learns kernel eigenfeatures from all proxy covariates and fits a ridge predictor on labeled data. We derive finite sample bounds showing that fast labeled sample rates are recovered when proxy perturbation is controlled and unlabeled proxy covariates are sufficiently abundant. We also show that distribution regression is a direct special case, with analogous guarantees when the finite bag size is large enough. Experiments show consistent gains over supervised and semi-supervised baselines, especially in low label regimes.

## 1. Introduction

Semi-supervised learning (SSL) seeks provable gains from combining a small labeled sample with a large unlabeled sample (van Engelen & Hoos, 2020; Zhu & Goldberg, 2009). Classical theory typically explains such gains through structure in the marginal covariate distribution, most notably the manifold and cluster (low-density separation) assumptions (Belkin & Niyogi, 2004; 2008; Belkin et al., 2006; Lafferty & Wasserman, 2007; Niyogi, 2013; Rigollet, 2007; Singh et al., 2008; Sinha & Belkin, 2009; Zhu et al., 2003). While these assumptions yield fast rates in idealized regimes, they are hard to validate from limited labels, and prior theory shows that unlabeled data need not improve performance unless the marginal covariate distribution is informative about the regression or decision function. Recent work also sharpens the picture for graph-based regularization by characterizing when Laplacian-based SSL does and does not help (Cabannes et al., 2021).

This paper studies a complementary obstacle that is ubiquitous in modern machine learning pipelines: the covariates available to the learner are often noisy proxies. In contemporary systems, including foundation-model pipelines, downstream predictors are trained on representations produced from massive unlabeled corpora, but these representations are imperfect, domain-dependent measurements of latent variables. Empirically, data curation and distributional mismatch can dominate performance even when unlabeled scale is enormous (Gadre et al., 2023). From a statistical viewpoint, this motivates SSL with noisy covariates: we observe labeled pairs $(\widehat{X}_i, Y_i)_{i=1}^n$ and additional unlabeled covariates $\widehat{X}_{n+1:N}$, where $\widehat{X}$ is a contaminated or proxy observation for an unobserved latent covariate $X$. The basic question is whether, and under what conditions, unlabeled proxy covariates improve generalization despite measurement error. Relatedly, recent theory on learning with noisy features underscores that "more data" and "more features" can interact in nonclassical ways even in supervised settings (Li et al., 2023), reinforcing the need for SSL analyses that explicitly model covariate noise.

Our approach is based on spectral methods and reproducing kernel Hilbert spaces. Using the unlabeled proxy covariates, we estimate leading eigenspaces of an empirical integral operator and fit the labeled regression on the resulting eigenfeatures, following the kernel projection viewpoint (Guo & Zhou, 2012; Ji et al., 2012). The technical core is a generalization analysis that tracks (i) approximation error from projecting onto a finite eigenspace, (ii) operator and eigenspace estimation error induced by finite $N$ and proxy perturbation, (iii) coefficient estimation error from $n$ labels, and (iv) ridge shrinkage. The bounds identify when sufficiently accurate unlabeled proxy covariates recover the oracle feature rate under polynomial eigenvalue decay.

### 1.1. Measurement Error and Distribution Regression

Measurement error (or errors-in-variables) models replace latent covariates $X$ by noisy observations $\widehat{X}$; while classical work is extensive, nonparametric measurement error with unknown error mechanisms remains challenging, and SSL

[1]Department of Statistics, Korea University, Seoul, Korea
[2]Department of Statistics, Seoul National University, Seoul, Korea. Correspondence to: Jisu Kim <jkim82133@snu.ac.kr>.

*Proceedings of the 43$^{rd}$ International Conference on Machine Learning*, Seoul, South Korea. PMLR 306, 2026. Copyright 2026 by the author(s).

versions of such problems are largely unexplored (Carroll et al., 2009; Schennach, 2016). Our framework treats SSL with noisy covariates as the primary object.

Distribution regression is a key special case, in which each covariate is a probability distribution $P$, observed only through a finite bag of samples, and one predicts $Y$ from $P$ (Oliva et al., 2013; Póczos et al., 2013; Szabó et al., 2016). This fits our framework by viewing $\widehat{P}$, constructed from finitely many samples, as a proxy for $P$. We develop theory for this case as well. Recent work on kernel based representations of distributions (Kachaiev & Recanatesi, 2024) further underscores the role of distribution valued covariates as a modern learning primitive.

## 1.2. Main results and contributions

We analyze a kernel projection semi-supervised estimator that first learns spectral features from all proxy covariates $\widehat{X}_{1:N}$ and then fits a regularized linear predictor using the labeled pairs $(\widehat{X}_i, Y_i)_{i=1}^n$. The main result is a finite sample excess risk bound that separates spectral truncation, proxy based operator error, labeled estimation, and ridge shrinkage. Under the stronger two sided polynomial decay condition $\lambda_s \asymp s^{-q}$, the bound yields the oracle feature rate $n^{-q/(q+1)}$ whenever the operator learning and proxy perturbation terms are negligible at the oracle truncation level. In the labeled only regime, when $N = n$, the additional operator learning term remains and its effect depends explicitly on the local eigengap sequence. We also show that distribution regression is a direct special case: the latent covariate is a distribution $P$ observed through a finite sample proxy $\widehat{P}$, and the generic proxy perturbation term becomes a bag size dependent error envelope.

Our contributions are fourfold. First, we develop finite sample generalization bounds for semi-supervised regression with noisy proxy covariates, allowing both well specified and misspecified regression functions and explicitly tracking the proxy perturbation floor. Second, the bounds characterize when unlabeled proxy covariates remove the statistical cost of learning the spectral representation, with explicit dependence on unlabeled sample size, local eigengap, proxy quality, and ridge regularization. Third, we show that distribution regression fits the same framework by viewing the empirical distribution $\widehat{P}$ as a proxy for a latent distribution $P$. Fourth, we provide implementable guarantees for validation based tuning and approximate eigendecomposition, showing that the same rates are preserved up to logarithmic and subspace approximation terms.

## 2. Set Up and Notation

Our goal is to establish a unified framework for semi-supervised regression with noisy proxy covariates. Let $\mathcal{U}$ be a compact metric space with metric $d$, and let $(U, Y) \sim \mathbb{P}$, where $U \in \mathcal{U}$ and $Y \in \mathbb{R}$. We observe $N$ covariate instances, of which only the first $n$ are labeled, with $n \leq N$. For labeled observations, the latent regression model is

$$Y_i = f(U_i) + \mu_i, \qquad i = 1, \ldots, n, \tag{1}$$

where $f : \mathcal{U} \to \mathbb{R}$ is unknown and the noise variables $\mu_i$ are independent, mean zero, and sub-Gaussian. We write $\tau_0^2 = \mathbb{E}[\mu_i^2]$. Throughout, we assume the proxy mechanism does not introduce outcome-noise bias, so that

$$\mathbb{E}[\mu_i \mid U_i, \widehat{U}_i] = 0.$$

This holds, for example, when the proxy $\widehat{U}_i$ is generated conditionally independently of $\mu_i$ given $U_i$.

The latent covariate $U_i$ is not observed. Instead, we observe proxy covariates

$$\widehat{U}_i \sim \mathsf{M}(\cdot \mid U_i), \qquad i = 1, \ldots, N,$$

generated from a possibly unknown measurement channel $\mathsf{M}$, with $\widehat{U}_i \in \mathcal{U}$. The observed sample is

$$\mathcal{D}_{\mathcal{U}}^{\mathrm{obs}} = \{(\widehat{U}_i, Y_i)\}_{i=1}^n \cup \{\widehat{U}_i\}_{i=n+1}^N.$$

This formulation covers settings where downstream predictors use noisy proxies of latent features, such as representations produced by large scale pretraining.

We next introduce the hypothesis space and target predictor. Let $\mathcal{H}_\kappa$ be a reproducing kernel Hilbert space (RKHS) on $\mathcal{U}$ with bounded Mercer kernel $\kappa : \mathcal{U} \times \mathcal{U} \to \mathbb{R}$. Define the best-in-class predictor

$$g \in \arg\min_{h \in \mathcal{H}_\kappa} \mathbb{E}\left[(f(U) - h(U))^2\right], \tag{2a}$$

$$\in \arg\min_{h \in \mathcal{H}_\kappa} \mathbb{E}\left[(Y - h(U))^2\right], \tag{2b}$$

$$\epsilon^2 = \min_{h \in \mathcal{H}_\kappa} \mathbb{E}\left[(f(U) - h(U))^2\right]. \tag{2c}$$

The two minimization problems in (2) have the same solution because $Y = f(U) + \mu$ and $\mathbb{E}[\mu \mid U] = 0$. Thus $g$ is the best-in-class latent predictor, and $\epsilon^2$ is the approximation error; $\epsilon^2 = 0$ in the well-specified case.

For $h \in \mathcal{H}_\kappa$, define the population kernel integral operator

$$\mathscr{L}(h)(\cdot) = \mathbb{E}[\kappa(U, \cdot)h(U)].$$

Given samples, define the latent empirical operator and the proxy empirical operator by

$$\mathscr{L}_N(h)(\cdot) = \frac{1}{N} \sum_{i=1}^N \kappa(U_i, \cdot)h(U_i),$$

$$\widehat{\mathscr{L}}_N(h)(\cdot) = \frac{1}{N} \sum_{i=1}^N \kappa(\widehat{U}_i, \cdot)h(\widehat{U}_i).$$

Our analysis controls $\|\widehat{\mathscr{L}}_N - \mathscr{L}\|$, which captures both finite sample operator error and proxy perturbation.

Let $\{(\hat{\lambda}_j, \hat{\phi}_j)\}_{j \geq 1}$ denote the eigenpairs of $\widehat{\mathscr{L}}_N$, ordered by nonincreasing eigenvalues, with eigenfunctions normalized in $\mathcal{H}_\kappa$. For $s \geq 1$, the estimator uses the leading eigenfunctions $\{\hat{\phi}_j\}_{j=1}^s$. We write $\| \cdot \|_{\mathcal{H}_\kappa}$ for the RKHS norm. Population eigenfunctions used in the analysis are normalized in $L_2(\mathbb{P}_U)$, and the relevant norm is written explicitly when needed.

The framework specializes to the two settings studied in this paper. First, for noisy Euclidean covariates analyzed in Section 3, we take $\mathcal{U} = \mathcal{X} \subset \mathbb{R}^d$, set $U = X$, and let $\widehat{U} = \widehat{X}$ be a noisy proxy of $X$. Second, for distribution regression, we take $\mathcal{U} = \mathcal{P}$, a metric space of probability measures with $d = D$, as specified in Section 4, set $U = P$, and let $\widehat{U} = \widehat{P}$ be a finite-sample estimator of $P$. In both cases, the observed semi-supervised sample has the form $\mathcal{D}_{\mathcal{U}}^{\mathrm{obs}}$, so the same algorithm and analysis apply after instantiating $(\mathcal{U}, d, \kappa, \mathsf{M})$.

## 3. SSL with Noisy Proxy Covariates

We study semi-supervised regression when the latent covariate $X$ is not observed and only a noisy proxy $\widehat{X}$ is available. The goal is to use the full proxy sample $\{\widehat{X}_i\}_{i=1}^N$ (labeled and unlabeled) to learn a low-dimensional spectral representation of the covariate geometry, and then fit the best-in-class predictor using the labeled responses.

### 3.1. Estimation

We now describe the estimation procedure explicitly. The method has two stages: (i) learn a data-dependent feature map from all proxy covariates, and (ii) fit a ridge-type regressor on the labeled data using these features.

**Stage 1: spectral feature extraction from proxy covariates.** Let $\mathcal{H}_\kappa$ be an RKHS on $\mathcal{X}$ with bounded Mercer kernel $\kappa : \mathcal{X} \times \mathcal{X} \to \mathbb{R}$. Define the population operator

$$\mathscr{L}(h)(\cdot) = \mathbb{E}\big[\kappa(X, \cdot)\, h(X)\big], \quad h \in \mathcal{H}_\kappa, \qquad (3)$$

and the proxy empirical operator based on the observed covariates $\widehat{X}_1, \ldots, \widehat{X}_N$,

$$\widehat{\mathscr{L}}_N(h)(\cdot) = \frac{1}{N} \sum_{i=1}^N \kappa(\widehat{X}_i, \cdot)\, h(\widehat{X}_i). \qquad (4)$$

Let $\{(\hat{\lambda}_j, \hat{\phi}_j)\}_{j \geq 1}$ be the eigenpairs of $\widehat{\mathscr{L}}_N$ ordered by nonincreasing $\hat{\lambda}_j$, with $\{\hat{\phi}_j\}$ orthonormal in $\mathcal{H}_\kappa$. For a chosen dimension $s \geq 1$, define the feature map

$$\hat{\Phi}_s(x) = (\hat{\phi}_1(x), \ldots, \hat{\phi}_s(x))^\top \in \mathbb{R}^s.$$

**Stage 2: ridge regression using labeled examples.** Using only the labeled pairs $\{(\widehat{X}_i, Y_i)\}_{i=1}^n$, fit

$$\hat{\gamma} \in \arg\min_{\gamma \in \mathbb{R}^s} \left\{ \frac{1}{n} \sum_{i=1}^n \big(Y_i - \langle \gamma, \hat{\Phi}_s(\widehat{X}_i)\rangle\big)^2 + \xi\|\gamma\|_2^2 \right\},$$
$$\hat{g}(\cdot) = \langle \hat{\gamma}, \hat{\Phi}_s(\cdot)\rangle, \qquad (5)$$

where $\xi \geq 0$ is a regularization parameter, and $\hat{g}$ denotes the fitted predictor. Thus, unlabeled and labeled proxy covariates jointly determine the feature map $\hat{\Phi}_s$, while labels enter only through the ridge fit for $\hat{\gamma}$.

**Kernel-matrix form.** For implementation, it is convenient to express the eigenfunctions of $\widehat{\mathscr{L}}_N$ in terms of the finite kernel matrix constructed from the proxy covariates. Let $\mathbf{K} \in \mathbb{R}^{N \times N}$ with entries $\mathbf{K}_{ij} = \kappa(\widehat{X}_i, \widehat{X}_j)$. If $(v_j, \sigma_j)$ are the leading eigenpairs of $\mathbf{K}$ (descending $\sigma_j$), then $\hat{\lambda}_j = \sigma_j / N$ and

$$\hat{\phi}_j(\cdot) = \sigma_j^{-1/2} \sum_{i=1}^N (v_j)_i\, \kappa(\widehat{X}_i, \cdot). \qquad (6)$$

In particular, $\hat{\Phi}_s(\widehat{X}_i)$ is obtained by evaluating (6) at $\widehat{X}_i$, and (5) becomes ordinary ridge regression in $\mathbb{R}^s$ with design vectors $\{\hat{\Phi}_s(\widehat{X}_i)\}_{i=1}^n$.

### 3.2. Generalization Error Bounds

We first state a minimal set of assumptions under which we derive finite-sample generalization bounds and fast rates.

- (A1) *Sub-Gaussian noise and controlled measurement error.* The regression noise $\mu_i$ is independent and identically distributed, mean-zero, sub-Gaussian. Moreover, for every $k \geq 1$ there exists $\theta > 0$ such that

$$\mathbb{E}\Big[D_2(\widehat{X}, X)^k \mid X\Big] \leq (\theta/2)^k.$$

- (A2) *Bounded Hölder kernel.* The kernel is bounded, $|\kappa(x, x)| \leq 1$, and $\kappa(x, \cdot)$ is Hölder in $\mathcal{H}_\kappa$: there exist $L_\kappa > 0$ and $0 < \beta_\kappa \leq 1$ such that

$$\|\kappa(\cdot, x) - \kappa(\cdot, x')\|_{\mathcal{H}_\kappa} \leq L_\kappa\, D_2(x, x')^{\beta_\kappa}, \quad \forall x, x' \in \mathcal{X}.$$

- (A3) *Polynomial eigenvalue decay.* Let $\{\lambda_j\}_{j \geq 1}$ be the eigenvalues of $\mathscr{L}$ in (3). There exist $a > 0$ and $q > 2$ such that $\lambda_j \leq a^2 j^{-q}$ for all $j \geq 1$.

- (A4) *Regular eigenfunctions.* Let $\{\phi_j\}_{j \geq 1}$ be the eigenfunctions of $\mathscr{L}$. There exists $M_\phi < \infty$ such that $\sup_{x \in \mathcal{X}} \sup_{j \geq 1} |\phi_j(x)| \leq M_\phi$. Moreover each $\phi_j$ is Hölder in $x$ with exponent $0 < \beta_\phi \leq 1$ and constants $\{L_{\phi_j}\}$ satisfying $\sum_{j \geq 1} \lambda_j L_{\phi_j}^2 < \infty$.

- (A5) *Sample size and eigengap condition.* Let

$$\Delta_s = \lambda_s - \lambda_{s+1}. \tag{7}$$

Assume $\Delta_s > 0$ and that $n$ and $N$ are large enough so that, with high probability,

$$\|\widehat{\mathscr{L}}_N - \mathscr{L}\|_{\mathrm{op}} \le \Delta_s/2, \tag{8}$$

which guarantees that the top-$s$ eigenspace of $\widehat{\mathscr{L}}_N$ is well aligned with that of $\mathscr{L}$ by the Davis–Kahan theorem (Yu et al., 2015). In particular, under (8) we have

$$\|\hat{\Pi}_s^{\mathcal{H}} - \Pi_s^{\mathcal{H}}\|_{\mathrm{op}} \lesssim \frac{\|\widehat{\mathscr{L}}_N - \mathscr{L}\|_{\mathrm{op}}}{\Delta_s}.$$

- (A6) *Target regularity.* For some fixed constant $R < \infty$, the best-in-class predictor satisfies

$$\|g\|_{\mathcal{H}_\kappa} \le R.$$

Assumption (A1) controls the proxy error $D_2(\widehat{X}, X)$ without requiring an additive measurement-error model. Assumption (A2) is standard in kernel learning: boundedness holds after rescaling, and Hölder continuity is satisfied by many kernels on compact domains. Assumption (A3) is a spectral-decay condition widely used in kernel ridge regression and spectral approximation; larger $q$ corresponds to faster eigen-decay and hence lower effective dimension. Assumption (A4) imposes bounded and Hölder eigenfunctions, ensuring stable evaluation and perturbation control. Assumption (A5) is the usual eigengap regime for identifying the leading $s$-dimensional subspace: (8) is exactly the condition under which Davis–Kahan perturbation bounds apply (Yu et al., 2015). Throughout, we write $r_s$ for any deterministic lower bound on the eigengap, $0 < r_s \le \Delta_s$; one may take $r_s = \Delta_s$. Assumption (A6) fixes the RKHS radius of the best-in-class predictor and is used to control spectral truncation, subspace perturbation, and the transfer from proxy evaluation at $\widehat{X}$ to latent evaluation at $X$. Overall, these assumptions align with those in prior spectral semi-supervised learning and kernel learning theory (e.g., Guo & Zhou 2012; Ji et al. 2012; Koltchinskii & Yuan 2010; Szabó et al. 2016), with Assumption (A1) capturing the additional proxy-covariate setting.

Concrete examples help contextualize some of these regularity conditions. Assumption (A1) covers, for example, bounded proxy perturbations or additive proxy noise with uniformly controlled moments on a compact covariate space. Assumption (A3) is standard for kernels whose integral operators have polynomial spectral decay, such as Sobolev or Matérn type kernels on compact domains. Assumption (A4) is natural for smooth bounded Mercer kernels on compact Euclidean domains, where leading eigenfunctions are often

bounded and Hölder continuous; for instance, in the periodic Fourier case on $[0, 1]$, sine and cosine eigenfunctions are uniformly bounded with Lipschitz constants growing polynomially in frequency, so $\sum_j \lambda_j L_{\phi_j}^2 < \infty$ holds under sufficiently fast spectral decay. Assumption (A5) covers finite rank or spiked models with separated leading components, and polynomial spectrum regimes in which the local gap $r_s$ is large enough relative to the operator perturbation. *Remark* 3.1 (Choice of truncation level). Assumption (A5) is stated for a fixed truncation level $s$ and the corresponding eigengap $\Delta_s$. The same perturbation argument can be extended to data adaptive choices of $s$, provided the selected index satisfies a corresponding separation condition. For example, one may choose $s$ from a candidate set and use $r_s$ as a deterministic lower bound on the relevant local eigengap. For clarity, we state the main results for a fixed $s$ and the standard gap $\Delta_s$.

Next, we analyze performance relative to the best-in-class predictor in the chosen RKHS. This target is defined on the latent covariate $X$, whereas the estimator is constructed from the observed proxy covariates $\widehat{X}$. The proxy perturbation is accounted for explicitly in the bound below. Define

$$g \in \arg\min_{h \in \mathcal{H}_\kappa} \mathbb{E}\big[(Y - h(X))^2\big],$$
$$\epsilon^2 = \min_{h \in \mathcal{H}_\kappa} \mathbb{E}\big[(f(X) - h(X))^2\big].$$

Since $Y = f(X) + \mu$ with $\mathbb{E}[\mu \mid X] = 0$, the cross term vanishes and

$$\mathbb{E}\big[(g(X) - Y)^2\big] = \mathbb{E}\big[(g(X) - f(X))^2\big] + \tau_0^2 = \epsilon^2 + \tau_0^2.$$

Define the excess error between $\widehat{g}$ and $g$ by

$$R_{s,n,N}^2 = \mathbb{E}\big[(\widehat{g}(X) - g(X))^2\big].$$

The next proposition provides an upper bound on $R_{s,n,N}^2$, which serves as the main technical estimate from which the regression and prediction error bounds follow. All proofs are deferred to the appendix.

**Theorem 3.2.** *Under Assumptions (A1)–(A6),*

$$\mathbb{E}\big[(\widehat{g}(X) - g(X))^2\big] \le R_{s,n,N}^2,$$

*where, for any eigengap lower bound $0 < r_s \le \Delta_s$,*

$$R_{s,n,N}^2 = \mathcal{O}_P\left(s^{-q} + \frac{1}{r_s^2}\left(\frac{1}{N} + C_{\beta',\theta}\right) + \frac{s}{n} + \frac{\xi}{\xi + \lambda_s^2}\right). \tag{9}$$

*Here $\beta' = \min\{\beta_\kappa, \beta_\phi\}$, and $C_{\beta',\theta}$ is a proxy perturbation constant, induced by Assumptions (A1)–(A4) and (A6), that accounts for both operator perturbation and proxy to latent evaluation transfer.*

Equation (9) makes the roles of $s$ (spectral truncation), $N$ (unlabeled sample size for operator learning), and $n$ (labeled

sample size for coefficient learning) explicit. Combining Theorem 3.2 with elementary $L_2$ arguments yields corresponding bounds for the regression and prediction risks.

**Theorem 3.3** (Regression error). *Under Assumptions (A1)–(A6),*

$$\mathbb{E}\big[(\widehat{g}(X) - f(X))^2\big] \leq \epsilon^2 + 2\epsilon R_{s,n,N} + R_{s,n,N}^2, \quad (10)$$

*where $R_{s,n,N}$ is defined in Theorem 3.2.*

**Theorem 3.4** (Prediction error). *Under the conditions of Theorem 3.3,*

$$\mathbb{E}\big[(\widehat{g}(\widehat{X}) - Y)^2\big] \leq \epsilon^2 + \tau_0^2 + 2\epsilon R_{s,n,N} + R_{s,n,N}^2, \quad (11)$$

*after enlarging the constant in $R_{s,n,N}^2$ to absorb the proxy evaluation transfer term.*

### 3.3. Optimal tuning and rates

We next discuss the rate implications of Theorem 3.2. Under $\lambda_s \asymp s^{-q}$, once the ridge shrinkage term in (9) is negligible, the oracle spectral tradeoff between $s^{-q}$ and $s/n$ yields a fast rate whenever the operator learning and proxy perturbation term is negligible at the oracle truncation level, as formalized below.

**Corollary 3.5.** *Suppose Assumptions (A1)–(A6) hold and $\lambda_s \asymp s^{-q}$. If, at $s \asymp n^{1/(q+1)}$,*

$$\frac{1}{r_s^2}\left\{\frac{1}{N} + C_{\beta',\theta}\right\} = o\left(s^{-q} + \frac{s}{n}\right) \quad (12)$$

*and*

$$\frac{\xi}{\xi + \lambda_s^2} = o\left(s^{-q} + \frac{s}{n}\right) \quad (13)$$

*hold, then choosing $s \asymp n^{1/(q+1)}$ yields*

$$R_{s,n,N}^2 = \mathcal{O}_P\left(n^{-q/(q+1)}\right).$$

*Consequently, Theorems 3.3 and 3.4 achieve the same rate for the excess terms beyond $\epsilon^2$ and $\epsilon^2 + \tau_0^2$, respectively.*

A sufficient condition for (13) is $\xi = o\{\lambda_s^2(s^{-q} + s/n)\}$ at $s \asymp n^{1/(q+1)}$; in particular, $\xi = 0$ is allowed when the unregularized fit is stable. Corollary 3.5 is a conditional oracle rate statement: it requires the combined operator and proxy perturbation to be negligible at the oracle truncation level. If $C_{\beta',\theta}$ is fixed away from zero, condition (12) generally fails as $s \to \infty$, and Theorem 3.2 should instead be read as a finite sample bias, variance, and perturbation bound with a proxy induced floor. The condition is therefore most relevant when the proxy error decreases with the sampling design, as in distribution regression with growing bag size.

To make the dependence on the eigengap explicit, suppose that the relevant local eigengap satisfies $r_s \gtrsim s^{-\alpha}$ for some

$\alpha \geq 0$. Since $\lambda_s \asymp s^{-q}$, the operator-learning term satisfies

$$\frac{1}{r_s^2}\left\{\frac{1}{N} + C_{\beta',\theta}\right\} \lesssim s^{q+2\alpha}\left\{\frac{1}{N} + C_{\beta',\theta}\right\}.$$

At the choice $s \asymp n^{1/(q+1)}$, condition (12) is implied by

$$\frac{1}{N} + C_{\beta',\theta} = o\left(n^{-(2q+2\alpha)/(q+1)}\right).$$

Equivalently, it suffices that

$$N \gg n^{(2q+2\alpha)/(q+1)}, \quad C_{\beta',\theta} = o\left(n^{-(2q+2\alpha)/(q+1)}\right).$$

In the labeled-only regime, $N = n$, the operator-learning term cannot in general be ignored. To isolate the effect of estimating the eigenspace from only labeled covariates, suppose for the moment that the proxy perturbation term is negligible. Under $r_s \gtrsim s^{-\alpha}$, the leading upper bound becomes

$$s^{-q} + \frac{s^{q+2\alpha}}{n} + \frac{s}{n}.$$

The term $s/n$ is lower order at the resulting optimizer, and balancing $s^{-q}$ with $s^{q+2\alpha}/n$ gives

$$s \asymp n^{1/(2q+2\alpha)}, \quad R_{s,n,n}^2 = \mathcal{O}_P\left(n^{-q/(2q+2\alpha)}\right).$$

Thus the labeled-only rate depends explicitly on the eigengap sequence and is not generally $n^{-1/2}$. For example, under the regular-spacing heuristic $r_s \asymp \Delta_s \asymp s^{-(q+1)}$, one has $\alpha = q + 1$, and the labeled-only rate implied by the present Davis–Kahan bound is

$$R_{s,n,n}^2 = \mathcal{O}_P\left(n^{-q/(4q+2)}\right).$$

Therefore, under the eigengap scaling $r_s \gtrsim s^{-\alpha}$, moving from the labeled-only regime $N = n$ to a regime with sufficiently many and sufficiently accurate unlabeled proxy covariates improves the bound from the eigengap-dependent labeled-only rate $n^{-q/(2q+2\alpha)}$ to the oracle feature rate $n^{-q/(q+1)}$. Equivalently, unlabeled proxy covariates can remove the statistical cost of learning the spectral representation, but the quantitative gain depends on the eigengap sequence and requires the proxy perturbation to vanish.

### 3.4. Practical considerations

This subsection states two consequences of the preceding theory that make the method implementable in practice: data-driven selection of the tuning parameters $(s, \xi)$ and stable use of approximate eigendecompositions.

#### 3.4.1. DATA-DRIVEN TUNING OF $s$ AND $\xi$

The oracle choices of $(s, \xi)$ in Corollary 3.5 depend on unknown spectral, eigengap, and proxy perturbation quantities.

We therefore choose $(s, \xi)$ by validation while still using all proxy covariates, labeled and unlabeled, to construct the eigenfeatures.

Split the labeled indices $\{1, \ldots, n\}$ into disjoint training and validation sets $\mathcal{I}_{\mathrm{tr}}$ and $\mathcal{I}_{\mathrm{va}}$, with sizes $n_{\mathrm{tr}}$ and $n_{\mathrm{va}}$. Using all proxy covariates $\{\widehat{X}_i\}_{i=1}^N$, compute $\widehat{\Phi}_s(\cdot)$ for $s \in \mathcal{S}$. For each $(s, \xi) \in \mathcal{S} \times \Xi$, fit $\widehat{g}_{s,\xi}$ by (5) using only $\mathcal{I}_{\mathrm{tr}}$, and choose

$$(\widehat{s}, \widehat{\xi}) \in \underset{(s,\xi) \in \mathcal{S} \times \Xi}{\arg\min} \frac{1}{n_{\mathrm{va}}} \sum_{i \in \mathcal{I}_{\mathrm{va}}} \left\{ Y_i - \widehat{g}_{s,\xi}(\widehat{X}_i) \right\}^2, \quad \widehat{g}_{\mathrm{sel}} = \widehat{g}_{\widehat{s}, \widehat{\xi}}.$$

The following theorem shows that this selector satisfies a finite-grid oracle inequality. We state it under the same stability and proxy-transfer conditions used in the proof of Theorem 3.2, uniformly over the candidate grid.

**Theorem 3.6.** *Assume Assumptions (A1)–(A6). Suppose that the finite-dimensional stability events used in Lemma A.8 hold uniformly over $(s, \xi) \in \mathcal{S} \times \Xi$, and that the squared validation losses satisfy the usual finite-class Bernstein condition for least-squares validation. Then, for every $\delta \in (0, 1)$, with probability at least $1 - \delta$,*

$$\mathbb{E}\left[ (\widehat{g}_{\mathrm{sel}}(X) - g(X))^2 \right] \lesssim \inf_{(s,\xi) \in \mathcal{S} \times \Xi} R^2_{s,n_{\mathrm{tr}},N} + \frac{\log(|\mathcal{S}||\Xi|/\delta)}{n_{\mathrm{va}}},$$

*where $R^2_{s,n_{\mathrm{tr}},N}$ denotes the bound in Theorem 3.2 with $n$ replaced by $n_{\mathrm{tr}}$. Consequently, if $n_{\mathrm{tr}} \asymp n_{\mathrm{va}} \asymp n$, the grid contains a pair satisfying the conditions of Corollary 3.5, and $|\mathcal{S}||\Xi|$ is polynomial in $n$, then $\widehat{g}_{\mathrm{sel}}$ attains the rate $n^{-q/(q+1)}$ up to logarithmic factors.*

Theorem 3.6 justifies ordinary validation over a modest grid. The result is conditional on the grid containing near-oracle choices and on the same operator, proxy, and ridge negligibility conditions required for Corollary 3.5; in particular, validation cannot remove a fixed proxy perturbation floor.

### 3.4.2. Approximate eigendecomposition

Computing the leading eigenpairs of the $N \times N$ kernel matrix can be a bottleneck when $N$ is large. In practice, one often constructs approximate eigenfeatures using methods such as Nyström or randomized eigensolvers (e.g., Drineas & Mahoney, 2005; Mahoney, 2011). We next show that the guarantees are stable under sufficiently accurate subspace approximation.

Let $\widetilde{\Phi}_s(\cdot) \in \mathbb{R}^s$ be an approximate feature map for the top-$s$ proxy eigenspace, and let $\widetilde{g}$ be the ridge estimator obtained by replacing $\widehat{\Phi}_s$ with $\widetilde{\Phi}_s$ in (5). Let $\widehat{\Pi}_s^{\mathcal{H}}$ and $\widetilde{\Pi}_s^{\mathcal{H}}$ denote the $\mathcal{H}_\kappa$-orthogonal projectors onto $\mathrm{span}\{\widehat{\phi}_1, \ldots, \widehat{\phi}_s\}$ and $\mathrm{span}\{\widetilde{\phi}_1, \ldots, \widetilde{\phi}_s\}$, respectively. We impose the following subspace accuracy condition on the approximate eigenspace.

- (A7) *Approximate eigenspace accuracy.* There exists $\varepsilon_{\mathrm{spec}} \geq 0$ such that $\|\widetilde{\Pi}_s^{\mathcal{H}} - \widehat{\Pi}_s^{\mathcal{H}}\|_{\mathrm{op}} \leq \varepsilon_{\mathrm{spec}}$.

Assumption (A7) is implied by standard subspace-accuracy guarantees. For example, for Nyström approximations, if the induced kernel matrix approximation error is controlled in operator norm, Davis–Kahan type perturbation bounds imply a corresponding bound on $\|\widetilde{\Pi}_s^{\mathcal{H}} - \widehat{\Pi}_s^{\mathcal{H}}\|_{\mathrm{op}}$, with the empirical eigengap entering the denominator. Randomized Lanczos and subspace iteration methods provide analogous subspace-error bounds, with $\varepsilon_{\mathrm{spec}}$ decreasing as the number of iterations increases.

**Theorem 3.7** (Approximate eigendecomposition). *Assume the conditions of Theorem 3.2, Assumption (A7), and the analogue of the stability event in Lemma A.8 for the approximate feature space. Then*

$$\mathbb{E}\left[ (\widetilde{g}(X) - g(X))^2 \right] \leq R^2_{s,n,N} + C\varepsilon_{\mathrm{spec}}^2,$$

*where $C$ depends only on fixed boundedness constants. Consequently, under the conditions of Corollary 3.5, if $s \asymp n^{1/(q+1)}$ and $\varepsilon_{\mathrm{spec}} = o\{n^{-q/(2q+2)}\}$, then $\widetilde{g}$ attains the same rate $n^{-q/(q+1)}$.*

Theorem 3.7 shows that approximate eigenfeature construction adds only a subspace-approximation term. Thus, the statistical rate is unchanged whenever $\varepsilon_{\mathrm{spec}}^2$ is smaller than the target statistical error.

In sum, Theorem 3.6 shows that the rate in Corollary 3.5 is attainable without oracle knowledge of $(s, \xi)$ when the grid contains near-oracle values, while Theorem 3.7 shows that scalable eigensolvers preserve the same rate provided the subspace approximation error is sufficiently small.

## 4. Application to Distribution Regression

This section applies the general framework of Section 3 to distribution regression, where each covariate is a probability distribution observed only through finitely many samples. The latent covariate is a distribution $P$, and the observed covariate is a proxy $\widehat{P}$ constructed from a finite bag of draws from $P$. Thus distribution regression is a structured noisy-covariate problem. The analysis below is a direct specialization of Section 3, with the bag size $m$ controlling the proxy perturbation level.

### 4.1. Setup

Let $\mathcal{Z} \subset \mathbb{R}^d$ be a compact domain and let $\mathcal{P}$ be a set of probability measures on $\mathcal{Z}$. We observe $N$ distribution-valued covariates $\{P_i\}_{i=1}^N$, of which only the first $n$ have labels. The labeled sample satisfies

$$Y_i = f(P_i) + \mu_i, \quad i = 1, \ldots, n, \tag{14}$$

where $f : \mathcal{P} \to \mathbb{R}$ is an unknown regression functional and $\{\mu_i\}$ are independent, mean-zero, sub-Gaussian noises with $\tau_0^2 = \mathbb{E}[\mu_i^2]$. Each distribution $P_i$ is not directly observed.

Instead, for each $i \in \{1, \ldots, N\}$, we observe a bag of samples

$$Z_{i1}, \ldots, Z_{im_i} \sim P_i \quad \text{i.i.d.,} \tag{15}$$

and construct a proxy $\widehat{P}_i$ from the bag $\overline{\mathcal{Z}}_i = \{Z_{i1}, \ldots, Z_{im_i}\}$. For simplicity, we take $m_i = m$ for all $i$; heterogeneous bag sizes can be handled by replacing $m$ with an effective bag size.

The observed semi-supervised data are

$$\mathcal{D}_{\mathcal{P}}^{\mathrm{obs}} = \{(\widehat{P}_i, Y_i)\}_{i=1}^n \cup \{\widehat{P}_i\}_{i=n+1}^N,$$

which has the same structure as Section 3 under the identification $X \equiv P$ and $\widehat{X} \equiv \widehat{P}$.

Let $\kappa : \mathcal{P} \times \mathcal{P} \to \mathbb{R}$ be a bounded positive definite kernel on distributions and let $\mathcal{H}_\kappa$ be its RKHS. Examples include kernels induced by distribution embeddings, such as mean embedding kernels (Smola et al., 2007), or kernels based on distances between distribution representations. Define the population operator

$$\mathscr{L}(h)(\cdot) = \mathbb{E}[\kappa(P, \cdot) h(P)],$$

and the proxy empirical operator

$$\widehat{\mathscr{L}}_N(h)(\cdot) = \frac{1}{N} \sum_{i=1}^N \kappa(\widehat{P}_i, \cdot) h(\widehat{P}_i).$$

Let $\{(\widehat{\lambda}_j, \widehat{\phi}_j)\}_{j \geq 1}$ be the eigenpairs of $\widehat{\mathscr{L}}_N$, ordered by nonincreasing eigenvalues and orthonormal in $\mathcal{H}_\kappa$. For $s \geq 1$, define

$$\widehat{\Phi}_s(P) = (\widehat{\phi}_1(P), \ldots, \widehat{\phi}_s(P))^\top.$$

Given labeled bags $\{(\widehat{P}_i, Y_i)\}_{i=1}^n$, fit

$$\widehat{\gamma} \in \underset{\gamma \in \mathbb{R}^s}{\arg\min} \left\{ \frac{1}{n} \sum_{i=1}^n \left( Y_i - \langle \gamma, \widehat{\Phi}_s(\widehat{P}_i) \rangle \right)^2 + \xi \|\gamma\|_2^2 \right\},$$

$$\widehat{g}(\cdot) = \langle \widehat{\gamma}, \widehat{\Phi}_s(\cdot) \rangle. \tag{16}$$

This is the same estimator as (5), applied to distribution-valued covariates.

## 4.2. Generalization bounds and rates

For analysis, we use the same structure as in Section 3, interpreted on the covariate space $\mathcal{P}$ with metric $D$. The only new ingredient is a bag-induced proxy error bound for $\widehat{P}$ relative to $P$. The following conditions parallel those in Section 3, but we restate them in the distribution-valued notation for completeness.

- (B1) *Sub-Gaussian noise and conditional exogeneity.* The noise variables $\mu_i$ in (14) are independent, mean-zero, and sub-Gaussian. Moreover,

$$\mathbb{E}[\mu_i \mid P_i, \widehat{P}_i] = 0.$$

This holds, for example, when the bag sampling mechanism is conditionally independent of $\mu_i$ given $P_i$.

- (B2) *Bounded Hölder kernel on distributions.* The kernel satisfies $|\kappa(P, P)| \leq 1$ and

$$\|\kappa(\cdot, P) - \kappa(\cdot, Q)\|_{\mathcal{H}_\kappa} \leq L_\kappa D(P, Q)^{\beta_\kappa}, \quad P, Q \in \mathcal{P},$$

for constants $L_\kappa > 0$ and $0 < \beta_\kappa \leq 1$.

- (B3) *Spectral decay and eigenfunction regularity.* The eigenvalues $\{\lambda_j\}_{j \geq 1}$ of $\mathscr{L}$ satisfy $\lambda_j \leq a^2 j^{-q}$ for some $a > 0$ and $q > 2$. The corresponding eigenfunctions are uniformly bounded and Hölder continuous on $\mathcal{P}$, as in Assumption (A4).

- (B4) *Bag-induced proxy error.* There exists a deterministic function $\rho(m)$ such that, for every $k \geq 1$,

$$\mathbb{E}\left[ D(P, \widehat{P})^k \mid P \right] \leq \rho(m)^k, \tag{17}$$

and $\rho(m) \to 0$ as $m \to \infty$. For density based estimators under smoothness conditions, a typical rate is $\rho(m) \asymp m^{-\beta/(2+d)}$ for some effective smoothness $\beta > 0$.

- (B5) *Sample size and eigengap condition.* Let $\Delta_s = \lambda_s - \lambda_{s+1}$. Assume $\Delta_s > 0$, and let $r_s$ be any deterministic lower bound satisfying $0 < r_s \leq \Delta_s$. Assume that $N$ and $m$ are large enough so that, with high probability,

$$\|\widehat{\mathscr{L}}_N - \mathscr{L}\|_{\mathrm{op}} \leq \Delta_s/2.$$

- (B6) *Target regularity.* For some fixed constant $R < \infty$, the best-in-class predictor satisfies

$$\|g\|_{\mathcal{H}_\kappa} \leq R.$$

Assumptions (B1)–(B6) are the distribution valued counterparts of the conditions in Section 3. The only additional ingredient is Assumption (B4), which controls the bag induced proxy error and determines the perturbation level in the distribution regression problem. Specifically, write

$$\beta' = \min\{\beta_\kappa, \beta_\phi\}, \quad a_m = \rho(m)^{\beta'} + \rho(m)^{2\beta'}.$$

The quantity $a_m$ is the bag induced proxy perturbation envelope. Since $\rho(m) \to 0$, it is of order $\rho(m)^{\beta'}$ for large $m$, and controls both the operator perturbation and the transfer from proxy evaluation at $\widehat{P}$ to latent evaluation at $P$.

Assumption (B4) holds for regular parametric families $\{P_\eta : \eta \in \Theta\}$ when an estimator $\widehat{\eta}$ is $m^{-1/2}$-consistent and the map $\eta \mapsto P_\eta$ is locally Lipschitz under the metric $D$, which gives $D(P_{\widehat{\eta}}, P_\eta) = O_P(m^{-1/2})$ and hence $\rho(m) \asymp m^{-1/2}$. It also covers smooth nonparametric density classes on compact domains, where standard density estimators satisfy $D(P, \widehat{P}) = O_P\{\rho(m)\}$ with the

usual smoothness dependent nonparametric rate. Assumption (B5) is the distribution valued counterpart of Assumption (A5); it holds in finite rank or spiked distributional feature models with separated leading components, and in polynomial spectrum regimes when $N$ and $m$ make the combined empirical and bag induced perturbation smaller than the local gap $r_s$.

Now, define

$$g \in \arg\min_{h \in \mathcal{H}_\kappa} \mathbb{E}\big[(Y - h(P))^2\big],$$
$$\epsilon^2 = \min_{h \in \mathcal{H}_\kappa} \mathbb{E}\big[(f(P) - h(P))^2\big],$$

and let

$$R_{s,n,N,m}^2 = \mathbb{E}\big[(\widehat{g}(P) - g(P))^2\big].$$

The next proposition specializes the main excess risk bound to distribution regression, with the generic proxy perturbation term replaced by the bag induced envelope $a_m$.

**Proposition 4.1.** *Assume Assumptions (A4) and (B1)–(B6). Then*

$$\mathbb{E}\big[(\widehat{g}(P) - g(P))^2\big] \leq R_{s,n,N,m}^2,$$

*where, for any eigengap lower bound $0 < r_s \leq \Delta_s$,*

$$R_{s,n,N,m}^2 = \mathcal{O}_P\left(s^{-q} + \frac{1}{r_s^2}\left\{\frac{1}{N} + a_m\right\} + \frac{s}{n} + \frac{\xi}{\xi + \lambda_s^2}\right). \tag{18}$$

The following theorem converts Proposition 4.1 into regression and prediction risk bounds for the latent distribution $P$ and its observed proxy $\widehat{P}$.

**Theorem 4.2.** *Assume Assumptions (A4) and (B1)–(B6). Then*

$$\mathbb{E}\big[(\widehat{g}(P) - f(P))^2\big] \leq \epsilon^2 + 2\epsilon R_{s,n,N,m} + R_{s,n,N,m}^2, \tag{19}$$

$$\mathbb{E}\left[(\widehat{g}(\widehat{P}) - Y)^2\right] \leq \epsilon^2 + \tau_0^2 + 2\epsilon R_{s,n,N,m} + R_{s,n,N,m}^2, \tag{20}$$

*after enlarging the constant in $R_{s,n,N,m}^2$ to absorb the same proxy evaluation transfer term.*

**Corollary 4.3.** *Assume Assumptions (A4), (B1)–(B6), and $\lambda_s \asymp s^{-q}$. If, at $s \asymp n^{1/(q+1)}$,*

$$\frac{1}{r_s^2}\left\{\frac{1}{N} + a_m\right\} = o\left(s^{-q} + \frac{s}{n}\right)$$

*and*

$$\frac{\xi}{\xi + \lambda_s^2} = o\left(s^{-q} + \frac{s}{n}\right),$$

*then choosing $s \asymp n^{1/(q+1)}$ yields*

$$R_{s,n,N,m}^2 = \mathcal{O}_P\left(n^{-q/(q+1)}\right).$$

*Consequently, the excess terms in (19) and (20) achieve the same rate.*

Corollary 4.3 makes explicit the three sample sizes governing distribution regression: $n$ labeled distributions determine the regression fit, $N$ total distributions determine the spectral representation, and $m$ samples per distribution determine the proxy perturbation $a_m$. The oracle feature rate is recovered only when $N$ is large enough and $m$ grows fast enough for the operator and proxy perturbation term to be negligible at the oracle truncation level.

## 5. Experiments

We evaluate the proposed estimator on two synthetic experiments and two real-world applications. The first synthetic experiment studies the noisy Euclidean covariate setting of Section 3, and the second studies the distribution regression setting of Section 4. For each task, we report normalized root mean squared error (RMSE). On real-world datasets, we compare against feasible supervised learning baselines (SL1–SL3) and semi-supervised learning baselines (SSL). Baseline definitions and tuning are provided in Appendices B.1 and B.2.

### 5.1. Synthetic noisy Euclidean covariates

We first evaluate the noisy covariate setting in Section 3. Latent covariates $X \in \mathbb{R}^{10}$ are generated from a smooth low-dimensional nonlinear structure, the response is $Y = f(X) + \epsilon$, and the learner observes only noisy proxies $\widehat{X} = X + \tau W$, where $W \sim N(0, I)$. We fix the number of labeled samples, vary the total number of proxy covariates $N$, and consider mild and strong proxy noise. Table 2 compares SSDR using all proxies, a labeled-only spectral baseline that learns features only from labeled proxies, and an oracle baseline using the true latent $X$. Increasing $N$ improves performance when proxy noise is mild; under stronger noise, gains are smaller and begin to saturate, consistent with the proxy perturbation floor in the theory. Details are provided in Appendix B.3.

| $\tau$ | $N$ | SSDR | Label-only | Oracle $X$ |
|---|---|---|---|---|
| .10 | 100 | .091 (.012) | .112 (.014) | .053 (.007) |
| .10 | 300 | .067 (.009) | .112 (.015) | .054 (.007) |
| .10 | 1000 | .057 (.008) | .113 (.014) | .053 (.008) |
| .40 | 100 | .176 (.019) | .198 (.022) | .054 (.007) |
| .40 | 300 | .153 (.017) | .197 (.021) | .053 (.008) |
| .40 | 1000 | .142 (.016) | .199 (.022) | .054 (.007) |

*Table 2.* Noisy Euclidean covariates: normalized RMSE, with standard deviations in parentheses.

| | | 5% | 10% | 15% | 20% | 25% | 30% | 35% | 40% |
|---|---|---|---|---|---|---|---|---|---|
| | | | | | $n/N \times 100(\%)$ | | | | |
| SL1 | SDM | 0.88 | 0.74 | 0.64 | 0.63 | 0.51 | 0.51 | 0.49 | 0.43 |
| SL2 | LASSO$_1$ | 0.90 | 0.69 | 0.65 | 0.68 | 0.66 | 0.64 | 0.63 | 0.62 |
| | SVR | 1.28 | 1.24 | 1.05 | 0.98 | 0.87 | 0.85 | 0.83 | 0.82 |
| | RF | 0.89 | 0.88 | 0.68 | 0.66 | 0.64 | 0.54 | 0.59 | 0.50 |
| | KRR | 0.98 | 0.99 | 0.85 | 0.70 | 0.77 | 0.74 | 0.68 | 0.66 |
| SL3 | LASSO$_2$ | 2.88 | 1.69 | 2.62 | 2.78 | 2.70 | 2.39 | 2.23 | 2.10 |
| | CNN | 2.52 | 1.84 | 1.44 | 1.36 | 1.45 | 1.41 | 1.34 | 1.04 |
| | RF | 1.88 | 1.69 | 1.52 | 1.68 | 1.01 | 1.20 | 0.99 | 1.03 |
| SSL | LapRLS | 1.01 | 0.79 | 0.70 | 0.63 | 0.56 | 0.54 | 0.51 | 0.48 |
| | E-CNN | 0.98 | 0.97 | 0.92 | 0.78 | 0.76 | 0.68 | 0.66 | 0.62 |
| | SSDR | **0.75** | **0.61** | **0.58** | **0.49** | **0.45** | **0.42** | **0.41** | **0.37** |

*Table 1.* Galaxy cluster mass prediction: normalized RMSE.

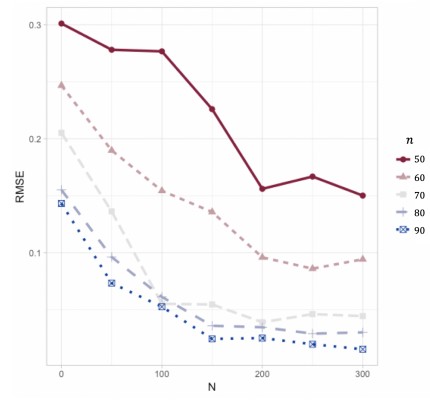

*Figure 1.* Normalized RMSE for predicting Beta skewness.

## 5.2. Synthetic distribution regression: predicting Beta skewness

This simulation setup follows Póczos et al. (2013). Each covariate instance is a Beta distribution $\mathrm{Beta}(a,b)$, the response is the skewness

$$f(a,b) = \frac{2(b-a)\sqrt{a+b-1}}{(a+b-2)\sqrt{ab}},$$

and the covariate is observed through a finite bag of samples. Figure 1 reports normalized RMSE as a function of the labeled and total sample sizes. Error decreases as $n$ increases and also decreases as $N$ increases, consistent with the role of unlabeled covariates in the bound. Details are in Appendix B.4.

## 5.3. Galaxy cluster mass from line-of-sight velocity distributions

We use the galaxy clusters dataset of Ntampaka et al. (2015) to predict halo mass from the distribution of line-of-sight velocities $v_{\mathrm{los}}$. Prior work often relies on low-dimensional summaries, whereas SSDR uses the full bag of $v_{\mathrm{los}}$ samples as a distributional covariate. We fix $N = 1632$ clusters and vary $n$ by subsampling labeled clusters. Table 1 shows that SSDR achieves the lowest normalized RMSE across label budgets. Details are in Appendix B.5.

## 5.4. Default risk from distributions of stock returns

We predict default probability from distributions of daily stock returns. We select the top 500 stocks by market capitalization in the Korea Stock Exchange and collect daily returns from 2010 to 2015. Labels are default probabilities obtained from FnGuide and Bloomberg. We vary the labeled fraction by randomly withholding labels while retaining all covariate bags. Table 3 shows that SSDR is competitive and typically best or tied for best, especially in the low-label regime. Details are in Appendix B.6.

| | | FnGuide | | Bloomberg | |
|---|---|---|---|---|---|
| | $n/N(\%)$ | 5 | 40 | 5 | 40 |
| SL1 | SDM | .12 | .05 | .09 | .07 |
| SL2 | LASSO$_1$ | .18 | .09 | .09 | .07 |
| | SVR | .15 | .09 | .10 | .09 |
| | RF | .11 | .08 | .08 | .07 |
| | KRR | .15 | .10 | .10 | .08 |
| SL3 | LASSO$_2$ | .51 | .39 | .18 | .16 |
| | CNN | .52 | .42 | .10 | .09 |
| | RF | .378 | .29 | .12 | .12 |
| SSL | LapRLS | .07 | **.04** | .08 | .07 |
| | E-CNN | **.06** | .06 | .07 | .07 |
| | SSDR | **.06** | .05 | **.07** | **.06** |

*Table 3.* Default risk prediction: normalized RMSE.

## 6. Discussion

This paper develops a unified theory for semi-supervised regression with noisy proxy covariates and shows that distribution regression is a direct specialization. The two stage estimator learns spectral features from all proxy covariates and fits ridge regression on labeled data. The generalization bounds identify when unlabeled proxies recover the oracle feature rate, with adaptive tuning and eigensolver stability guarantees.

The main limitation is that the fast rate requires a stable leading eigenspace and sufficiently accurate proxies. More unlabeled proxy covariates reduce representation learning error, but they do not remove a fixed proxy error floor. In distribution regression, this appears through the bag size dependent proxy term. For large $N \times N$ kernels, Nyström or randomized eigensolvers are often required (Section 3.4.2). Future work includes adaptive diagnostics for spectral and proxy quality, weaker eigengap analyses, and sharper proxy error bounds for modern learned representations.

## Impact Statement

This paper studies semi-supervised regression when covariates are observed through noisy proxies. The techniques are broadly applicable across domains, and any downstream societal impact will depend on the specific application context and data; we do not identify risks that are uniquely induced by the methodology beyond those common to standard predictive modeling.

## Acknowledgements

This work was supported in part by grants from the National Research Foundation of Korea funded by the Korean government (MSIT) (Nos. RS-2022-NR068754, RS-2024-00335008, and RS-2025-24534596), by the Samsung Science and Technology Foundation under Project Number SSTF-BA2502-01, and by Korea University Grant K2612051. This work was also supported in part by the New Faculty Startup Fund from Seoul National University (No. 0670-20240073), and by Institute of Information & communications Technology Planning & Evaluation (IITP) grant funded by the Korea government(MSIT) (No. RS-2021-II211343, Artificial Intelligence Graduate School Program (Seoul National University)).

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

# APPENDIX

# A. Proof

## A.1. Proofs for Section 3

Throughout, $c, C > 0$ denote finite constants that may change from line to line and may depend on fixed problem parameters (e.g., $a, L_\kappa, M_\phi, \theta, \beta_\kappa, \beta_\phi$), but never on $(n, N, s, \xi)$. We write $A \lesssim B$ for $A \leq CB$ for some constant $C < \infty$.

### A.1.1. NOTATION AND BASIC IDENTITIES

Let $(X, Y) \sim \mathbb{P}$ with

$$Y = f(X) + \mu, \qquad \mathbb{E}[\mu \mid X, \widehat{X}] = 0,$$

where $\mu$ is sub-Gaussian as in Assumption (A1). The conditional mean-zero condition holds, for example, when the proxy channel generating $\widehat{X}$ is conditionally independent of $\mu$ given $X$. Let $\mathcal{H}_\kappa$ be the reproducing kernel Hilbert space associated with $\kappa$, and let $\mathscr{L}$ be the population operator in (3). Let $\{(\lambda_j, \phi_j)\}_{j \geq 1}$ be the eigenpairs of $\mathscr{L}$ in $L_2(\mathbb{P}_X)$, ordered as $\lambda_1 \geq \lambda_2 \geq \cdots$, so that $\{\phi_j\}_{j \geq 1}$ is orthonormal in $L_2(\mathbb{P}_X)$ and

$$\mathscr{L}\phi_j = \lambda_j \phi_j.$$

For $h \in \mathcal{H}_\kappa$, the standard kernel integral operator identity gives

$$\|h\|_{L_2(\mathbb{P}_X)}^2 = \langle h, \mathscr{L}h \rangle_{\mathcal{H}_\kappa}. \tag{21}$$

Hence, since $\|\mathscr{L}\|_{\mathrm{op}} = \lambda_1 \leq a^2$ under Assumption (A3),

$$\|h\|_{L_2(\mathbb{P}_X)}^2 \leq \lambda_1 \|h\|_{\mathcal{H}_\kappa}^2 \leq a^2 \|h\|_{\mathcal{H}_\kappa}^2. \tag{22}$$

Let

$$g \in \underset{h \in \mathcal{H}_\kappa}{\arg\min} \, \mathbb{E}\big[(Y - h(X))^2\big], \qquad \epsilon^2 = \min_{h \in \mathcal{H}_\kappa} \mathbb{E}\big[(f(X) - h(X))^2\big].$$

Since $Y = f(X) + \mu$ and $\mathbb{E}[\mu \mid X] = 0$,

$$\mathbb{E}\big[(g(X) - Y)^2\big] = \mathbb{E}\big[(g(X) - f(X))^2\big] + \tau_0^2 = \epsilon^2 + \tau_0^2.$$

**Proxy perturbation convention.** Throughout the proof, $\beta' = \min\{\beta_\kappa, \beta_\phi\}$, and $C_{\beta',\theta}$ denotes a deterministic proxy perturbation envelope satisfying

$$\mathbb{E}\Big[D_2(\widehat{X}, X)^{\beta'}\Big] + \mathbb{E}\Big[D_2(\widehat{X}, X)^{2\beta'}\Big] + \Big\{\mathbb{E}\Big[D_2(\widehat{X}, X)^{2\beta'}\Big]\Big\}^{1/2} + 2\left(\theta \cdot 2^{\frac{(1-\beta')}{\beta'} + \frac{1}{2}}\right)^{\beta'} \leq C_{\beta',\theta}.$$

Under Assumption (A1), such an envelope exists by Jensen's inequality. We use the same notation after enlarging constants, so that $C_{\beta',\theta}$ controls both the operator perturbation caused by observing $\widehat{X}$ instead of $X$ and the risk transfer from proxy evaluation to latent evaluation.

### A.1.2. AUXILIARY LEMMAS

**Lemma A.1** (Bounded proxy spectral features). *Under Assumption (A2), for every $s \geq 1$,*

$$\|\widehat{\Phi}_s(\widehat{X})\|_2 \leq 1$$

*almost surely.*

*Proof.* Because $\{\widehat{\phi}_j\}_{j \geq 1}$ is orthonormal in $\mathcal{H}_\kappa$,

$$\|\widehat{\Phi}_s(\widehat{X})\|_2^2 = \sum_{j=1}^s \widehat{\phi}_j(\widehat{X})^2 = \sum_{j=1}^s \langle \widehat{\phi}_j, \kappa(\widehat{X}, \cdot) \rangle_{\mathcal{H}_\kappa}^2 \leq \|\kappa(\widehat{X}, \cdot)\|_{\mathcal{H}_\kappa}^2 = \kappa(\widehat{X}, \widehat{X}) \leq 1,$$

where the inequality follows from Bessel's inequality. $\square$

**Lemma A.2** (Approximation by top-$s$ eigenfunctions). *Assume Assumptions (A3) and (A6). Let $\Pi_s$ be the $L_2(\mathbb{P}_X)$-orthogonal projection onto* $\mathrm{span}\{\phi_1, \ldots, \phi_s\}$ *and set* $g_s = \Pi_s g$. *Then*

$$\|g - g_s\|^2_{L_2(\mathbb{P}_X)} \leq \lambda_{s+1} \|g\|^2_{\mathcal{H}_\kappa} \lesssim s^{-q}.$$

*Proof.* Using the usual spectral representation of $\mathcal{H}_\kappa$ with respect to the orthonormal basis $\{\sqrt{\lambda_j}\phi_j\}$:

$$g = \sum_{j \geq 1} b_j \sqrt{\lambda_j} \phi_j, \qquad \text{with} \qquad \sum_{j \geq 1} b_j^2 = \|g\|^2_{\mathcal{H}_\kappa}.$$

Then

$$g_s = \sum_{j \leq s} b_j \sqrt{\lambda_j} \phi_j$$

and therefore

$$\|g - g_s\|^2_{L_2(\mathbb{P}_X)} = \sum_{j > s} b_j^2 \lambda_j \leq \lambda_{s+1} \sum_{j > s} b_j^2 \leq \lambda_{s+1} \|g\|^2_{\mathcal{H}_\kappa}.$$

The conclusion follows from Assumptions (A3) and (A6). $\qquad\qquad\qquad\qquad\qquad\qquad\qquad\square$

**Lemma A.3.** *Under Assumption (A1), for all $\beta$ such that $0 < \beta \leq 1$, with probability $1 - \delta$ we have*

$$\frac{1}{N} \sum_{i=1}^N D_2^\beta(\hat{X}_i, X_i) \leq C_{\beta, \theta} \left( 1 + \frac{\log(2/\delta)}{\sqrt{N}} \right).$$

*Proof.* First, for $\forall i$ and $\forall l \geq 1$, note that the $2l$-th moment of $D_2^\beta(\hat{X}_i, X_i) - \mathbb{E}\left[D_2^\beta(\hat{X}_i, X_i)\right]$ is bounded as

$$\mathbb{E}\left[\left(D_2^\beta(\hat{X}_i, X_i) - \mathbb{E}\left[D_2^\beta(\hat{X}_i, X_i)\right]\right)^{2l}\right] \leq \mathbb{E}\left[\left(D_2^\beta(\hat{X}_i, X_i) - \mathbb{E}\left[D_2^\beta(\hat{X}_i, X_i)\right]\right)^{2l}\right]$$

$$\leq 2^{2l-1}\left(\mathbb{E}\left[D_2^{2l\beta}(\hat{X}_i, X_i)\right] + \left(\mathbb{E}\left[D_2^\beta(\hat{X}_i, X_i)\right]\right)^{2l}\right)$$

$$\leq 2^{2l}\mathbb{E}\left[D_2^{2l\beta}(\hat{X}_i, X_i)\right],$$

where the last inequality follows by Jensen's inequality.

Next, by Assumption (A1) it follows

$$\mathbb{E}\left[D_2^k(\hat{X}_i, X_i)\right] = \mathbb{E}\left\{\mathbb{E}\left[D_2^k(\hat{X}_i, X_i) \mid X_i\right]\right\}$$

$$\leq \left(\frac{\theta}{2}\right)^k$$

$$\leq \Gamma\left(\frac{k}{2} + 1\right)\left(\frac{\theta}{2}\right)^k$$

for every $k > 0$ and some global constant $\theta$, and thus

$$\mathbb{E}\left[\left(D_2^\beta(\hat{X}_i, X_i) - \mathbb{E}\left[D_2^\beta(\hat{X}_i, X_i)\right]\right)^{2l}\right] \leq (l\beta)! \left(2^{\frac{l(1-\beta)}{l\beta}}\theta\right)^{2l\beta}$$

$$\leq \frac{(2l)!}{2^l l!}\left(2^{\frac{(1-\beta)}{\beta}}\theta\right)^{2l\beta},$$

where the last inequality follows simply by the fact that $\beta \leq 1$. Hence, $D_2^\beta(\hat{X}_i, X_i) - \mathbb{E}\left[D_2^\beta(\hat{X}_i, X_i)\right]$ is sub-Gaussian with parameter $\frac{1}{2}\Theta$, where we let $\Theta \equiv 2\left(\theta \cdot 2^{\frac{(1-\beta)}{\beta} + \frac{1}{2}}\right)^\beta$.

Now by Hoeffding's inequality, we have

$$\mathbb{P}\left(\left|\frac{1}{N}\sum_{i=1}^{N}D_2^{\beta}(\hat{X}_i,X_i)-\mathbb{E}\left[D_2^{\beta}(\hat{X}_i,X_i)\right]\right|\geq t\right)\leq 2\exp\left(-\frac{Nt^2}{\Theta}\right),$$

and therefore with probability $1-\delta$,

$$\left|\frac{1}{N}\sum_{i=1}^{N}D_2^{\beta}(\hat{X}_i,X_i)-\mathbb{E}\left[D_2^{\beta}(\hat{X}_i,X_i)\right]\right|\leq\frac{\Theta^{1/2}\log(2/\delta)}{\sqrt{N}}.$$

Hence with probability $1-\delta$,

$$\frac{1}{N}\sum_{i=1}^{N}D_2^{\beta}(\hat{X}_i,X_i)=\mathbb{E}\left[D_2^{\beta}(\hat{X}_i,X_i)\right]+\left|\frac{1}{N}\sum_{i=1}^{N}D_2^{\beta}(\hat{X}_i,X_i)-\mathbb{E}\left[D_2^{\beta}(\hat{X}_i,X_i)\right]\right|$$

$$\leq\mathbb{E}\left[D_2^{\beta}(\hat{X}_i,X_i)\right]+\frac{\Theta^{1/2}\log(2/\delta)}{\sqrt{N}}$$

$$\leq C_{\beta,\theta}^{1/2}\left(1+\frac{\log(2/\delta)}{\sqrt{N}}\right).$$

$\square$

**Lemma A.4** (Proxy operator deviation)**.** *Assume Assumptions (A1)–(A4). Let*

$$\mathscr{L}_N(h)=\frac{1}{N}\sum_{i=1}^{N}\kappa(X_i,\cdot)h(X_i),\qquad\widehat{\mathscr{L}}_N(h)=\frac{1}{N}\sum_{i=1}^{N}\kappa(\hat{X}_i,\cdot)h(\hat{X}_i).$$

*Then*

$$\|\widehat{\mathscr{L}}_N-\mathscr{L}\|_{HS}=\mathcal{O}_P\left(N^{-1/2}+C_{\beta',\theta}^{1/2}\right),\qquad\|\widehat{\mathscr{L}}_N-\mathscr{L}\|_{HS}^2=\mathcal{O}_P\left(\frac{1}{N}+C_{\beta',\theta}\right).$$

*Proof.* Decompose

$$\widehat{\mathscr{L}}_N-\mathscr{L}=(\mathscr{L}_N-\mathscr{L})+(\widehat{\mathscr{L}}_N-\mathscr{L}_N).$$

For the sampling term, define

$$T_i:h\mapsto\kappa(X_i,\cdot)h(X_i).$$

Then $\mathscr{L}_N=N^{-1}\sum_{i=1}^{N}T_i$ and $\mathbb{E}[T_i]=\mathscr{L}$. Then since $\left\{\sqrt{\lambda_j}\phi_j\right\}$ is the orthonormal basis in $\mathcal{H}_\kappa$,

$$\|T_i\|_{HS}^2=\sum_{j=1}^{\infty}\left\|T_i\sqrt{\lambda_j}\phi_j\right\|^2=\sum_{j=1}^{\infty}\lambda_j\phi_j^2(X_i)\|\kappa_{X_i}\|_{\mathcal{H}_\kappa}^2$$

$$=\|\kappa_{X_i}\|_{\mathcal{H}_\kappa}^2\sum_{j=1}^{\infty}\lambda_j\phi_j^2(X_i).$$

Then $\|\kappa_{X_i}\|_{\mathcal{H}_\kappa}^2=|\kappa(X_i,X_i)|\leq 1$ holds under (A2), $\sum_{j=1}^{\infty}\lambda_j\leq\frac{qa^2}{q-1}\leq 2a^2$ holds under (A3), and $\phi_j^2(X_i)\leq M_\phi^2$ holds under (A4), and hence $\|T_i\|_{HS}\leq\sqrt{2}aM_\phi$ is uniformly bounded. Note that $\mathbb{E}[T_i]=\mathscr{L}$, hence from the concentration in the Hilbert space of Hilbert–Schmidt operators ((Smale & Zhou, 2009), Proposition 1), with probability $1-\delta$,

$$\left\|\mathscr{L}-\hat{\mathscr{L}}_N\right\|_{HS}\leq\frac{4\sqrt{2}aM_\phi\log(2/\delta)}{\sqrt{N}}. \tag{23}$$

For the proxy perturbation term, define

$$\hat{T}_i:h\mapsto\kappa(\hat{X}_i,\cdot)h(\hat{X}_i).$$

Then

$$\widehat{\mathscr{L}}_N - \mathscr{L}_N = \frac{1}{N}\sum_{i=1}^N (\widehat{T}_i - T_i).$$

Then similarly as above,

$$\left\|\widehat{T}_i - T_i\right\|_{HS}^2 = \sum_{j=1}^\infty \left\|(\widehat{T}_i - T_i)\sqrt{\lambda_j}\phi_j\right\|_{\mathcal{H}_\kappa}^2 = \sum_{j=1}^\infty \lambda_j \left\|\phi_j(\hat{X}_i)\kappa_{\hat{X}_i} - \phi_j(X_i)\kappa_{X_i}\right\|_{\mathcal{H}_\kappa}^2$$

$$\le \sum_{j=1}^\infty \lambda_j \left(\left\|\phi_j(X_i)\kappa_{X_i} - \phi_j(\hat{X}_i)\kappa_{X_i}\right\|_{\mathcal{H}_\kappa} + \left\|\phi_j(\hat{X}_i)\kappa_{X_i} - \phi_j(\hat{X}_i)\kappa_{\hat{X}_i}\right\|_{\mathcal{H}_\kappa}\right)^2.$$

Then $\left|\phi_j(X_i) - \phi_j(\hat{X}_i)\right| \le L_\phi D_2(X_i, \hat{X}_i)^{\beta_\phi}$ under (A4) and $\left\|\kappa_{X_i} - \kappa_{\hat{X}_i}\right\|_{\mathcal{H}_\kappa} \le L_\kappa D_2(X_i, \hat{X}_i)^{\beta_\kappa}$ under (A2), hence with $\|\kappa_{X_i}\|_{\mathcal{H}_\kappa}^2 \le 1$ under (A2), $\sum_{j=1}^\infty \lambda_j \le 2a^2$ under (A3), and $\phi_j^2(X_i) \le M_\phi^2$ under (A4),

$$\left\|\widehat{T}_i - T_i\right\|_{HS} \le \sqrt{2}a(L_\phi + L_\kappa M_\phi)(D_2(X_i, \hat{X}_i)^{\beta_\phi} + D_2(X_i, \hat{X}_i)^{\beta_\kappa}).$$

And hence

$$\left\|\widehat{\mathscr{L}}_N - \mathscr{L}_N\right\|_{HS} \le \sqrt{2}a(L_\phi + L_\kappa M_\phi)\frac{1}{N}\sum_{i=1}^N (D_2(X_i, \hat{X}_i)^{\beta_\phi} + D_2(X_i, \hat{X}_i)^{\beta_\kappa}).$$

Then under (A1), applying Lemma A.3 gives that, with probability $1 - 2\delta$,

$$\left\|\widehat{\mathscr{L}}_N - \mathscr{L}_N\right\|_{HS} \le 2\sqrt{2}a(L_\phi + L_\kappa M_\phi)C_{\beta',\theta}^{1/2}\left(1 + \frac{\log(2/\delta)}{\sqrt{N}}\right), \tag{24}$$

i.e., $\left\|\widehat{\mathscr{L}}_N - \mathscr{L}_N\right\|_{HS}$ is bounded by a constant of order $C_{\beta',\theta}^{1/2}$.

Combining the two displays and squaring gives that, with probability $1 - 3\delta$,

$$\left\|\mathscr{L} - \tilde{\mathscr{L}}_N\right\|_{HS} \le \left\|\mathscr{L} - \hat{\mathscr{L}}_N\right\|_{HS} + \left\|\widehat{\mathscr{L}}_N - \mathscr{L}_N\right\|_{HS}$$

$$\le \frac{4\sqrt{2}aM_\phi \log(2/\delta)}{\sqrt{N}} + 2\sqrt{2}a(L_\phi + L_\kappa M_\phi)C_{\beta',\theta}^{1/2}\left(1 + \frac{\log(2/\delta)}{\sqrt{N}}\right),$$

which proves the claim. $\qquad\square$

Hereafter, let

$$\Delta_s = \lambda_s - \lambda_{s+1}.$$

Throughout the appendix, $r_s$ denotes any deterministic lower bound on the eigengap:

$$0 < r_s \le \Delta_s.$$

In particular, one may take $r_s = \Delta_s$. The next lemma converts the operator deviation bound into a perturbation bound for the leading spectral subspace learned from the proxy covariates.

**Lemma A.5** (Davis–Kahan for the top-$s$ subspace). *Let $\Pi_s^{\mathcal{H}}$ be the $\mathcal{H}_\kappa$-orthogonal projector onto*

$$\mathrm{span}\{\sqrt{\lambda_1}\phi_1, \ldots, \sqrt{\lambda_s}\phi_s\},$$

*and let $\widehat{\Pi}_s^{\mathcal{H}}$ be the $\mathcal{H}_\kappa$-orthogonal projector onto*

$$\mathrm{span}\{\widehat{\phi}_1, \ldots, \widehat{\phi}_s\}.$$

*If $\|\widehat{\mathscr{L}}_N - \mathscr{L}\|_{\mathrm{op}} \le \Delta_s/2$, then*

$$\|\widehat{\Pi}_s^{\mathcal{H}} - \Pi_s^{\mathcal{H}}\|_{\mathrm{op}} \le \frac{2\|\widehat{\mathscr{L}}_N - \mathscr{L}\|_{\mathrm{op}}}{\Delta_s} \le \frac{2\|\widehat{\mathscr{L}}_N - \mathscr{L}\|_{HS}}{r_s}.$$

*Proof.* This is the standard Davis–Kahan sin-$\Theta$ bound for compact self-adjoint operators; see, for example, Yu et al. (2015, Theorem 2). The last inequality uses $\|\cdot\|_{\mathrm{op}} \le \|\cdot\|_{HS}$ and $r_s \le \Delta_s$. $\qquad\square$

**Lemma A.6** (Subspace perturbation). *Assume Assumptions (A1)–(A6). Let*

$$g_s^{\mathcal{H}} = \Pi_s^{\mathcal{H}} g, \qquad \widehat{g}_s^{\mathcal{H}} = \widehat{\Pi}_s^{\mathcal{H}} g.$$

*Then*

$$\|\widehat{g}_s^{\mathcal{H}} - g_s^{\mathcal{H}}\|_{L_2(\mathbb{P}_X)}^2 = \mathcal{O}_P\left\{ \frac{1}{r_s^2} \left( \frac{1}{N} + C_{\beta',\theta} \right) \right\}.$$

*Proof.* By (22),

$$\|\widehat{g}_s^{\mathcal{H}} - g_s^{\mathcal{H}}\|_{L_2(\mathbb{P}_X)}^2 \le a^2 \|\widehat{g}_s^{\mathcal{H}} - g_s^{\mathcal{H}}\|_{\mathcal{H}_\kappa}^2.$$

Since

$$\widehat{g}_s^{\mathcal{H}} - g_s^{\mathcal{H}} = (\widehat{\Pi}_s^{\mathcal{H}} - \Pi_s^{\mathcal{H}}) g,$$

we have $\left\| \widehat{g}_s^{\mathcal{H}} - g_s^{\mathcal{H}} \right\|_{\mathcal{H}_\kappa}^2$ being upper bounded as

$$\left\| \widehat{g}_s^{\mathcal{H}} - g_s^{\mathcal{H}} \right\|_{\mathcal{H}_\kappa}^2 = \left\| (\widehat{\Pi}_s^{\mathcal{H}} - \Pi_s^{\mathcal{H}})(g) \right\|_{\mathcal{H}_\kappa}^2$$

$$\le \left\| \widehat{\Pi}_s^{\mathcal{H}} - \Pi_s^{\mathcal{H}} \right\|_{\mathrm{op}}^2 \|g\|_{\mathcal{H}_\kappa}^2.$$

And then applying Lemma A.5 (with $r_s$ as a lower bound on $\Delta_s$) gives that

$$\left\| \widehat{\Pi}_s^{\mathcal{H}} - \Pi_s^{\mathcal{H}} \right\|_{\mathrm{op}}^2 \le \frac{4 \left\| \widehat{\mathscr{L}}_N - \mathscr{L} \right\|_{HS}^2}{r_s^2}.$$

Then we apply Lemma A.4 to get

$$\left\| \widehat{\Pi}_s^{\mathcal{H}} - \Pi_s^{\mathcal{H}} \right\|_{\mathrm{op}}^2 = \mathcal{O}_P\left( r_s^{-2}(N^{-1} + C_{\beta',\theta}) \right).$$

And then with $\|g\|_{\mathcal{H}_\kappa}^2 \le R^2$ from (A6), $\left\| \widehat{g}_s^{\mathcal{H}} - g_s^{\mathcal{H}} \right\|_{\mathcal{H}_\kappa}^2$ is upper bounded as

$$\left\| \widehat{g}_s^{\mathcal{H}} - g_s^{\mathcal{H}} \right\|_{\mathcal{H}_\kappa}^2 = \mathcal{O}_P\left( r_s^{-2}(N^{-1} + C_{\beta',\theta}) \right),$$

and $\left\| \widehat{g}_s^{\mathcal{H}} - g_s^{\mathcal{H}} \right\|_{L_2(\mathbb{P}_X)}^2$ is correspondingly bounded as

$$\left\| \widehat{g}_s^{\mathcal{H}} - g_s^{\mathcal{H}} \right\|_{L_2(\mathbb{P}_X)}^2 = \mathcal{O}_P\left( r_s^{-2}(N^{-1} + C_{\beta',\theta}) \right).$$

$\qquad\square$

**Lemma A.7** (Proxy evaluation transfer). *Assume Assumptions (A1) and (A2). Let*

$$\mathcal{G}_B = \{ h \in \mathcal{H}_\kappa : \|h\|_{\mathcal{H}_\kappa} \le B \},$$

*where $B < \infty$. Then, uniformly over $h \in \mathcal{G}_B$,*

$$\mathbb{E}\left[ (h(\widehat{X}) - h(X))^2 \right] \lesssim B^2 C_{\beta',\theta}.$$

*Moreover,*

$$\left| \mathbb{E}\left[ (Y - h(\widehat{X}))^2 \right] - \mathbb{E}\left[ (Y - h(X))^2 \right] \right| \lesssim (1 + B^2) C_{\beta',\theta}.$$

*Proof.* For the first claim, for $h \in \mathcal{G}_B$, the reproducing property and Assumption (A2) imply

$$
\begin{aligned}
|h(\hat{X}) - h(X)| &= \left| \langle h, \kappa(\cdot, \hat{X}) \rangle - \langle h, \kappa(\cdot, X) \rangle \right| \\
&= \left| \langle h, \kappa(\cdot, \hat{X}) - \kappa(\cdot, X) \rangle \right| \\
&\leq \|h\|_{\mathcal{H}_\kappa} \|\kappa_{\hat{X}} - \kappa_X\|_{\mathcal{H}_\kappa} \\
&\leq B \|\kappa_{\hat{X}} - \kappa_X\|_{\mathcal{H}_\kappa} \\
&\leq B L_\kappa D_2(\hat{X}, X)^{\beta_\kappa},
\end{aligned}
$$

where the first equality is from the reproducing property, the first inequality comes from Cauchy-Schwarz, and the last inequality is due to Assumption (A2). Notice that $B L_\kappa$ is a multiplication of predefined constants. Hence this with further bounding $\mathbb{E}\left[ D_2(\hat{X}, X)^{2\beta_\kappa} \right] \leq C_{\beta', \theta}$ under (A1), finally we obtain

$$
\begin{aligned}
\mathbb{E}\left[ \left( h(\hat{X}) - h(X) \right) \right] &\leq B^2 L_\kappa^2 \mathbb{E}\left[ D_2(\hat{X}, X)^{2\beta_\kappa} \right] \\
&\leq B^2 L_\kappa^2 C_{\beta', \theta}.
\end{aligned}
\tag{25}
$$

For the second claim, write

$$
(Y - h(\hat{X}))^2 - (Y - h(X))^2 = \{h(X) - h(\hat{X})\}\{2Y - h(\hat{X}) - h(X)\}.
$$

Since $\kappa(x, x) \leq 1$, every $h \in \mathcal{G}_B$ satisfies

$$
|h(x)| \leq \|h\|_{\mathcal{H}_\kappa} \sqrt{\kappa(x, x)} \leq B.
$$

By Cauchy–Schwarz and the sub-Gaussian moment bound for $Y$,

$$
\left| \mathbb{E}\left[ (Y - h(\hat{X}))^2 - (Y - h(X))^2 \right] \right| \lesssim \left\{ \mathbb{E}\left[ (h(\hat{X}) - h(X))^2 \right] \right\}^{1/2} (1 + B).
$$

By the proxy perturbation convention, the right-hand side is bounded by a constant multiple of $(1 + B^2) C_{\beta', \theta}$ after enlarging the envelope $C_{\beta', \theta}$. $\qquad \square$

**Lemma A.8** (Estimation in the learned feature space). *Condition on $\hat{X}_{1:N}$ and fix $s$. Let*

$$
\hat{\mathcal{F}}_s = \operatorname{span}\{\hat{\phi}_1, \ldots, \hat{\phi}_s\}.
$$

*Let $\hat{g}_{s,\xi}$ be the estimator in* (5). *Suppose that the finite-dimensional stability event*

$$
\mathcal{E}_{s,n} = \left\{ \|\hat{g}_{s,\xi}\|_{\mathcal{H}_\kappa}^2 \vee \|\hat{\Pi}_s^{\mathcal{H}} g\|_{\mathcal{H}_\kappa}^2 \leq B_s^2 \right\}
$$

*holds, where*

$$
B_s^2 \lesssim \frac{1}{r_s^2}.
$$

*Then, on $\mathcal{E}_{s,n}$,*

$$
\mathbb{E}\left[ (\hat{g}_{s,\xi}(X) - g(X))^2 \mid \hat{X}_{1:N} \right] \lesssim \inf_{\substack{h \in \hat{\mathcal{F}}_s \\ \|h\|_{\mathcal{H}_\kappa} \leq B_s}} \mathbb{E}\left[ (h(X) - g(X))^2 \mid \hat{X}_{1:N} \right] + \frac{s}{n} + \frac{\xi}{\xi + \lambda_s^2} + \frac{1}{r_s^2} C_{\beta', \theta}.
$$

*Proof.* By Lemma A.1, $\|\hat{\Phi}_s(\hat{X})\|_2 \leq 1$. Conditional on $\hat{X}_{1:N}$, the second stage is therefore an ordinary $s$-dimensional ridge regression problem with bounded design. On the event $\mathcal{E}_{s,n}$, the standard finite-dimensional ridge oracle inequality gives

$$
\mathbb{E}\left[ (\hat{g}_{s,\xi}(\hat{X}) - Y)^2 \mid \hat{X}_{1:N} \right] \leq \inf_{\substack{h \in \hat{\mathcal{F}}_s \\ \|h\|_{\mathcal{H}_\kappa} \leq B_s}} \mathbb{E}\left[ (h(\hat{X}) - Y)^2 \mid \hat{X}_{1:N} \right] + \mathcal{O}_P\left( \frac{s}{n} + \frac{\xi}{\xi + \lambda_s^2} \right).
$$

The term $s/n$ is the finite-dimensional coefficient estimation error, and the term $\xi/(\xi+\lambda_s^2)$ is the ridge shrinkage contribution.

By Lemma A.7, both sides may be transferred from $\widehat{X}$ to $X$ at cost

$$(1 + B_s^2) C_{\beta',\theta} \lesssim \frac{1}{r_s^2} C_{\beta',\theta}.$$

Finally, since $g$ is the $L_2(\mathbb{P}_X)$-projection of $f$ onto $\mathcal{H}_\kappa$, for every $h \in \mathcal{H}_\kappa$,

$$\mathbb{E}\big[(Y - h(X))^2\big] - \mathbb{E}\big[(Y - g(X))^2\big] = \mathbb{E}\big[(h(X) - g(X))^2\big].$$

Combining the preceding displays proves the result. □

### A.1.3. PROOF OF THEOREM 3.2

*Proof of Theorem 3.2.* We work on the intersection of the eigengap event in Assumption (A5) and the finite-dimensional stability event $\mathcal{E}_{s,n}$ in Lemma A.8. Since these events hold with probability tending to one under the stated conditions, the resulting bound is an $\mathcal{O}_P$ statement.

By Lemma A.8,

$$\mathbb{E}\big[(\widehat{g}(X) - g(X))^2\big] \lesssim \inf_{\substack{h \in \widehat{\mathcal{F}}_s \\ \|h\|_{\mathcal{H}_\kappa} \leq B_s}} \mathbb{E}\big[(h(X) - g(X))^2\big] + \frac{s}{n} + \frac{\xi}{\xi + \lambda_s^2} + \frac{1}{r_s^2} C_{\beta',\theta}.$$

Take

$$h = \widehat{g}_s^{\mathcal{H}} = \widehat{\Pi}_s^{\mathcal{H}} g.$$

On $\mathcal{E}_{s,n}$, this function belongs to $\widehat{\mathcal{F}}_s$ and satisfies $\|\widehat{g}_s^{\mathcal{H}}\|_{\mathcal{H}_\kappa} \leq B_s$. Hence

$$\inf_{\substack{h \in \widehat{\mathcal{F}}_s \\ \|h\|_{\mathcal{H}_\kappa} \leq B_s}} \mathbb{E}\big[(h(X) - g(X))^2\big] \leq \|\widehat{g}_s^{\mathcal{H}} - g\|_{L_2(\mathbb{P}_X)}^2.$$

Using the triangle inequality,

$$\|\widehat{g}_s^{\mathcal{H}} - g\|_{L_2(\mathbb{P}_X)}^2 \lesssim \|g - \Pi_s g\|_{L_2(\mathbb{P}_X)}^2 + \|\Pi_s^{\mathcal{H}} g - \widehat{\Pi}_s^{\mathcal{H}} g\|_{L_2(\mathbb{P}_X)}^2.$$

Lemma A.2 gives

$$\|g - \Pi_s g\|_{L_2(\mathbb{P}_X)}^2 \lesssim s^{-q}.$$

Lemma A.6 gives

$$\|\Pi_s^{\mathcal{H}} g - \widehat{\Pi}_s^{\mathcal{H}} g\|_{L_2(\mathbb{P}_X)}^2 = \mathcal{O}_P\bigg\{\frac{1}{r_s^2}\bigg(\frac{1}{N} + C_{\beta',\theta}\bigg)\bigg\}.$$

Combining the preceding displays yields

$$\mathbb{E}\big[(\widehat{g}(X) - g(X))^2\big] = \mathcal{O}_P\bigg(s^{-q} + \frac{1}{r_s^2}\bigg\{\frac{1}{N} + C_{\beta',\theta}\bigg\} + \frac{s}{n} + \frac{\xi}{\xi + \lambda_s^2}\bigg),$$

which proves (9). □

### A.1.4. PROOF OF THEOREM 3.3

*Proof of Theorem 3.3.* By the triangle inequality in $L_2(\mathbb{P}_X)$,

$$\|\widehat{g} - f\|_{L_2(\mathbb{P}_X)} \leq \|\widehat{g} - g\|_{L_2(\mathbb{P}_X)} + \|g - f\|_{L_2(\mathbb{P}_X)}.$$

Since $\|g - f\|_{L_2(\mathbb{P}_X)}^2 = \epsilon^2$ and Theorem 3.2 gives

$$\|\widehat{g} - g\|_{L_2(\mathbb{P}_X)}^2 = \mathbb{E}\big[(\widehat{g}(X) - g(X))^2\big] \leq R_{s,n,N}^2,$$

we obtain

$$\mathbb{E}\big[(\widehat{g}(X) - f(X))^2\big] = \|\widehat{g} - f\|_{L_2(\mathbb{P}_X)}^2 \leq (R_{s,n,N} + \epsilon)^2.$$

Expanding the right-hand side gives

$$\mathbb{E}\big[(\widehat{g}(X) - f(X))^2\big] \leq \epsilon^2 + 2\epsilon R_{s,n,N} + R_{s,n,N}^2,$$

which proves (10). □

### A.1.5. PROOF OF THEOREM 3.4

*Proof of Theorem 3.4.* We work on the same high-probability stability event used in Theorem 3.2. On this event, Lemma A.7 applied to $h = \widehat{g}$ gives

$$\left| \mathbb{E}\left[ (Y - \widehat{g}(\widehat{X}))^2 \right] - \mathbb{E}\left[ (Y - \widehat{g}(X))^2 \right] \right| \lesssim \frac{1}{r_s^2} C_{\beta', \theta}.$$

This term is of the same order as the proxy perturbation contribution in $R_{s,n,N}^2$ and is absorbed by enlarging its constant. Thus,

$$\mathbb{E}\left[ (Y - \widehat{g}(\widehat{X}))^2 \right] \leq \mathbb{E}\left[ (Y - \widehat{g}(X))^2 \right] + \text{a term absorbed into } R_{s,n,N}^2.$$

Since $Y = f(X) + \mu$ and $\mathbb{E}[\mu \mid X] = 0$,

$$\mathbb{E}\left[ (Y - \widehat{g}(X))^2 \right] = \mathbb{E}\left[ (f(X) - \widehat{g}(X))^2 \right] + \tau_0^2.$$

Applying Theorem 3.3 yields

$$\mathbb{E}\left[ (Y - \widehat{g}(\widehat{X}))^2 \right] \leq \epsilon^2 + \tau_0^2 + 2\epsilon R_{s,n,N} + R_{s,n,N}^2,$$

after enlarging the constant in $R_{s,n,N}^2$. This proves (11). □

### A.1.6. PROOF OF COROLLARY 3.5

*Proof of Corollary 3.5.* Let

$$s_n \asymp n^{1/(q+1)}.$$

By assumptions (12) and (13),

$$\frac{1}{r_{s_n}^2} \left\{ \frac{1}{N} + C_{\beta', \theta} \right\} = o\left( s_n^{-q} + \frac{s_n}{n} \right)$$

and

$$\frac{\xi}{\xi + \lambda_{s_n}^2} = o\left( s_n^{-q} + \frac{s_n}{n} \right).$$

Therefore Theorem 3.2 gives

$$R_{s_n,n,N}^2 = \mathcal{O}_P\left( s_n^{-q} + \frac{s_n}{n} \right).$$

Since

$$s_n^{-q} \asymp n^{-q/(q+1)} \qquad \text{and} \qquad \frac{s_n}{n} \asymp n^{-q/(q+1)},$$

we obtain

$$R_{s_n,n,N}^2 = \mathcal{O}_P\left( n^{-q/(q+1)} \right).$$

The claims for the regression and prediction risks follow from Theorems 3.3 and 3.4. □

### A.1.7. PROOF OF THEOREM 3.6

*Proof of Theorem 3.6.* Condition on the full proxy covariate sample $\widehat{X}_{1:N}$ and on the training labels used to construct the candidate predictors. Then the finite set

$$\mathcal{G} = \{\widehat{g}_{s,\xi} : (s, \xi) \in \mathcal{S} \times \Xi\}$$

is fixed with respect to the validation labels. Let

$$\mathcal{R}_{\widehat{X}}(h) = \mathbb{E}\left[ (Y - h(\widehat{X}))^2 \mid \widehat{X}_{1:N}, \mathcal{I}_{\text{tr}} \right]$$

and

$$\widehat{\mathcal{R}}_{\text{va}}(h) = \frac{1}{n_{\text{va}}} \sum_{i \in \mathcal{I}_{\text{va}}} \left\{ Y_i - h(\widehat{X}_i) \right\}^2.$$

By the finite-class Bernstein inequality for squared loss under bounded predictions and sub-Gaussian noise, with probability at least $1 - \delta$,

$$\mathcal{R}_{\widehat{X}}(\widehat{g}_{\mathrm{sel}}) \leq \inf_{h \in \mathcal{G}} \mathcal{R}_{\widehat{X}}(h) + C \frac{\log(|\mathcal{S}||\Xi|/\delta)}{n_{\mathrm{va}}}.$$

Equivalently,

$$\mathcal{R}_{\widehat{X}}(\widehat{g}_{\mathrm{sel}}) - \mathcal{R}_{\widehat{X}}(g) \leq \inf_{(s,\xi) \in \mathcal{S} \times \Xi} \left\{ \mathcal{R}_{\widehat{X}}(\widehat{g}_{s,\xi}) - \mathcal{R}_{\widehat{X}}(g) \right\} + C \frac{\log(|\mathcal{S}||\Xi|/\delta)}{n_{\mathrm{va}}}.$$

By the proxy transfer lemma, uniformly over the stable candidate class,

$$\mathcal{R}_{\widehat{X}}(h) - \mathcal{R}_{\widehat{X}}(g) = \mathbb{E}\left[ (h(X) - g(X))^2 \mid \widehat{X}_{1:N}, \mathcal{I}_{\mathrm{tr}} \right] + \text{a term absorbed into } R^2_{s,n_{\mathrm{tr}},N}.$$

Therefore,

$$\mathbb{E}\left[ (\widehat{g}_{\mathrm{sel}}(X) - g(X))^2 \right] \lesssim \inf_{(s,\xi) \in \mathcal{S} \times \Xi} \mathbb{E}\left[ (\widehat{g}_{s,\xi}(X) - g(X))^2 \right] + \frac{\log(|\mathcal{S}||\Xi|/\delta)}{n_{\mathrm{va}}},$$

after absorbing the uniform proxy-transfer contribution into the corresponding $R^2_{s,n_{\mathrm{tr}},N}$ terms. Applying Theorem 3.2 with the training size $n_{\mathrm{tr}}$ gives

$$\mathbb{E}\left[ (\widehat{g}_{\mathrm{sel}}(X) - g(X))^2 \right] \lesssim \inf_{(s,\xi) \in \mathcal{S} \times \Xi} R^2_{s,n_{\mathrm{tr}},N} + \frac{\log(|\mathcal{S}||\Xi|/\delta)}{n_{\mathrm{va}}},$$

which proves the oracle inequality.

If $n_{\mathrm{tr}} \asymp n_{\mathrm{va}} \asymp n$, the grid contains a pair satisfying the conditions of Corollary 3.5, and $|\mathcal{S}||\Xi|$ is polynomial in $n$, the validation remainder is logarithmic over $n$ and is smaller than the oracle rate up to logarithmic factors. The final rate claim follows. $\square$

### A.1.8. PROOF OF THEOREM 3.7

*Proof of Theorem 3.7.* Let

$$\widetilde{\mathcal{F}}_s = \mathrm{span}\{\widetilde{\phi}_1, \ldots, \widetilde{\phi}_s\}.$$

The proof follows the oracle-approximation argument used for Theorem 3.2, with $\widetilde{\mathcal{F}}_s$ in place of $\widehat{\mathcal{F}}_s$.

By the approximate-feature analogue of Lemma A.8,

$$\mathbb{E}\left[ (\widetilde{g}(X) - g(X))^2 \right] \lesssim \inf_{\substack{h \in \widetilde{\mathcal{F}}_s \\ \|h\|_{\mathcal{H}_\kappa} \leq B_s}} \mathbb{E}\left[ (h(X) - g(X))^2 \right] + \frac{s}{n} + \frac{\xi}{\xi + \lambda_s^2} + \frac{1}{\lambda_s r_s^2} C_{\beta',\theta}.$$

Take

$$h = \widetilde{\Pi}_s^{\mathcal{H}} \widehat{\Pi}_s^{\mathcal{H}} g.$$

Then $h \in \widetilde{\mathcal{F}}_s$, and by Assumption (A7),

$$\|h - \widehat{\Pi}_s^{\mathcal{H}} g\|_{\mathcal{H}_\kappa} = \|(\widetilde{\Pi}_s^{\mathcal{H}} - \widehat{\Pi}_s^{\mathcal{H}}) \widehat{\Pi}_s^{\mathcal{H}} g\|_{\mathcal{H}_\kappa} \leq \varepsilon_{\mathrm{spec}} \|g\|_{\mathcal{H}_\kappa}.$$

Using (22), Assumption (A6), and the triangle inequality,

$$\|h - g\|^2_{L_2(\mathbb{P}_X)} \lesssim \|\widehat{\Pi}_s^{\mathcal{H}} g - g\|^2_{L_2(\mathbb{P}_X)} + \varepsilon^2_{\mathrm{spec}}.$$

The first term is exactly the approximation and learned-subspace perturbation term controlled in the proof of Theorem 3.2. Therefore,

$$\mathbb{E}\left[ (\widetilde{g}(X) - g(X))^2 \right] \leq R^2_{s,n,N} + C \varepsilon^2_{\mathrm{spec}},$$

after enlarging constants. If the conditions of Corollary 3.5 hold, $s \asymp n^{1/(q+1)}$, and $\varepsilon_{\mathrm{spec}} = o\{n^{-q/(2q+2)}\}$, then $\varepsilon^2_{\mathrm{spec}} = o\{n^{-q/(q+1)}\}$, so the rate is unchanged. $\square$

## A.2. Proofs for Section 4

Throughout, $c, C > 0$ denote finite constants that may change from line to line and may depend only on fixed problem parameters, but never on $(n, N, s, \xi, m)$. We write $A \lesssim B$ for $A \leq CB$. Let

$$\beta' = \min\{\beta_\kappa, \beta_\phi\}, \qquad a_m = \rho(m)^{\beta'} + \rho(m)^{2\beta'}.$$

By Assumption (B4), $a_m$ is a proxy perturbation envelope in the sense that the first and second order proxy moments needed below are bounded by constants times $a_m$, after enlarging constants.

### A.2.1. AUXILIARY LEMMAS

**Lemma A.9** (Bounded proxy spectral features). *Under Assumption (B2), for every $s \geq 1$,*

$$\|\widehat{\Phi}_s(\widehat{P})\|_2 \leq 1$$

*almost surely.*

*Proof.* Because $\{\widehat{\phi}_j\}_{j \geq 1}$ is orthonormal in $\mathcal{H}_\kappa$,

$$\|\widehat{\Phi}_s(\widehat{P})\|_2^2 = \sum_{j=1}^{s} \widehat{\phi}_j(\widehat{P})^2 = \sum_{j=1}^{s} \langle \widehat{\phi}_j, \kappa(\widehat{P}, \cdot) \rangle_{\mathcal{H}_\kappa}^2 \leq \|\kappa(\widehat{P}, \cdot)\|_{\mathcal{H}_\kappa}^2 = \kappa(\widehat{P}, \widehat{P}) \leq 1.$$

$\square$

**Lemma A.10** (Spectral approximation). *Assume Assumptions (B3) and (B6). Let $\Pi_s$ be the $L_2(\mathbb{P}_P)$-orthogonal projection onto $\mathrm{span}\{\phi_1, \ldots, \phi_s\}$ and set $g_s = \Pi_s g$. Then*

$$\|g - g_s\|_{L_2(\mathbb{P}_P)}^2 \leq \lambda_{s+1} \|g\|_{\mathcal{H}_\kappa}^2 \lesssim s^{-q}.$$

*Proof.* The proof is identical to Lemma A.2, replacing $X$ by $P$. $\square$

**Lemma A.11.** *Under Assumption (B4), for all $\beta \geq 0$, with probability $1 - \delta$,*

$$\frac{1}{N} \sum_{i=1}^{N} D_2^\beta(\hat{P}_i, P_i) \leq \rho(m)^\beta \left(1 + \frac{\log(2/\delta)}{\sqrt{N}}\right).$$

*Proof.* Given $\beta$, similarly in Lemma A.3, for all $l \geq 1$ $2l$-moment of $D_2^\beta(\hat{P}_i, P_i) - \mathbb{E}\left[D_2^\beta(\hat{P}_i, P_i)\right]$ can be bounded as

$$\mathbb{E}\left[\left(D_2^\beta(\hat{P}_i, P_i) - \mathbb{E}\left[D_2^\beta(\hat{P}_i, P_i)\right]\right)^{2l}\right] \leq 2^{2l} \mathbb{E}\left[D_2^{2l\beta}(\hat{P}_i, P_i)\right].$$

Next, by Assumption (B4) it follows

$$\mathbb{E}\left[D_2^k(\hat{P}_i, P_i)\right] = \mathbb{E}\left\{\mathbb{E}\left[D_2^k(\hat{P}_i, P_i) \mid P_i\right]\right\}$$
$$\leq \rho(m)^k,$$

for every $k > 0$, and thus

$$\mathbb{E}\left[\left(D_2^\beta(\hat{p}_i, p_i) - \mathbb{E}\left[D_2^\beta(\hat{p}_i, p_i)\right]\right)^{2l}\right] \leq \rho(m)^{2l\beta}$$
$$\leq \frac{(2l)!}{2^l l!} \rho(m)^{2l\beta}.$$

This implies that $D_2^\beta(\hat{p}_i, p_i) - \mathbb{E}\left[D_2^\beta(\hat{p}_i, p_i)\right]$ is sub-Gaussian with its parameter $\left(\sqrt{2}\rho(m)\right)^{2\beta}$. Hence by applying Hoeffding's,

$$\mathbb{P}\left(\left|\frac{1}{N}\sum_{i=1}^N D_2^\beta(\hat{p}_i, p_i) - \mathbb{E}\left[D_2^\beta(\hat{p}_i, p_i)\right]\right| \geq t\right) \leq 2\exp\left(-\frac{Nt^2}{\left(\sqrt{2}\rho(m)\right)^{2\beta}}\right),$$

and therefore with probability $1 - \delta$,

$$\left|\frac{1}{N}\sum_{i=1}^N D_2^\beta(\hat{p}_i, p_i) - \mathbb{E}\left[D_2^\beta(\hat{p}_i, p_i)\right]\right| \leq \frac{\left(\sqrt{2}\rho(m)\right)^\beta \log(2/\delta)}{\sqrt{N}}.$$

Hence with probability $1 - \delta$,

$$\frac{1}{N}\sum_{i=1}^N D_2^\beta(\hat{P}_i, p_i) = \mathbb{E}\left[D_2^\beta(\hat{P}_i, P_i)\right] + \left|\frac{1}{N}\sum_{i=1}^N D_2^\beta(\hat{P}_i, P_i) - \mathbb{E}\left[D_2^\beta(\hat{P}_i, P_i)\right]\right|$$

$$\leq \rho(m)^\beta\left(1 + \frac{2^{\beta/2}\log(2/\delta)}{\sqrt{N}}\right).$$

$\square$

**Lemma A.12** (Proxy operator deviation). *Assume Assumptions (A4) and (B2)–(B4). Then*

$$\|\widehat{\mathscr{L}}_N - \mathscr{L}\|_{HS} = \mathcal{O}_P\left(N^{-1/2} + a_m^{1/2}\right), \qquad \|\widehat{\mathscr{L}}_N - \mathscr{L}\|_{HS}^2 = \mathcal{O}_P\left(\frac{1}{N} + a_m\right).$$

*Proof.* We proceed similar to Lemma A.4. Decompose

$$\widehat{\mathscr{L}}_N - \mathscr{L} = (\mathscr{L}_N - \mathscr{L}) + (\widehat{\mathscr{L}}_N - \mathscr{L}_N).$$

For the sampling term, define

$$T_i : h \mapsto \kappa(P_i, \cdot)h(P_i).$$

Then $\mathscr{L}_N = \frac{1}{N}\sum_i T_i$ and $\mathbb{E}[T_i] = \mathscr{L}$. Then since $\{\sqrt{\lambda_j}\phi_j\}$ is the orthonormal basis in $\mathcal{H}_\kappa$,

$$\|T_i\|_{HS}^2 = \sum_{j=1}^\infty \left\|T_i\sqrt{\lambda_j}\phi_j\right\|^2 = \sum_{j=1}^\infty \lambda_j\phi_j^2(P_i)\|\kappa_{P_i}\|_{\mathcal{H}_\kappa}^2$$

$$= \|\kappa_{P_i}\|_{\mathcal{H}_\kappa}^2 \sum_{j=1}^\infty \lambda_j\phi_j^2(P_i).$$

Then $\|\kappa_{P_i}\|_{\mathcal{H}_\kappa}^2 = |\kappa(P_i, P_i)| \leq 1$ holds under (B2), $\sum_{j=1}^\infty \lambda_j \leq \frac{qa^2}{q-1} \leq 2a^2$ holds under (B3), and $\phi_j^2(P_i) \leq M_\phi^2$ holds under (B4), and hence $\|T_i\|_{HS} \leq \sqrt{2}aM_\phi$ is uniformly bounded. Note that $\mathbb{E}[T_i] = \mathscr{L}$, hence from the concentration in the Hilbert space of Hilbert–Schmidt operators ((Smale & Zhou, 2009), Proposition 1), with probability $1 - \delta$,

$$\left\|\mathscr{L} - \hat{\mathscr{L}}_N\right\|_{HS} \leq \frac{4\sqrt{2}aM_\phi \log(2/\delta)}{\sqrt{N}}. \tag{26}$$

For the bag-induced proxy term, define

$$\widehat{T}_i : h \mapsto \kappa(\widehat{P}_i, \cdot)h(\widehat{P}_i).$$

Then

$$\widehat{\mathscr{L}}_N - \mathscr{L}_N = \frac{1}{N}\sum_i(\widehat{T}_i - T_i).$$

Then similarly as above,

$$\left\|\widehat{T}_i - T_i\right\|_{HS}^2 = \sum_{j=1}^{\infty} \left\|(\widehat{T}_i - T_i)\sqrt{\lambda_j}\phi_j\right\|_{\mathcal{H}_\kappa}^2 = \sum_{j=1}^{\infty} \lambda_j \left\|\phi_j(\hat{P}_i)\kappa_{\hat{P}_i} - \phi_j(P_i)\kappa_{P_i}\right\|_{\mathcal{H}_\kappa}^2$$

$$\leq \sum_{j=1}^{\infty} \lambda_j \left(\left\|\phi_j(P_i)\kappa_{P_i} - \phi_j(\hat{P}_i)\kappa_{P_i}\right\|_{\mathcal{H}_\kappa} + \left\|\phi_j(\hat{P}_i)\kappa_{P_i} - \phi_j(\hat{P}_i)\kappa_{\hat{P}_i}\right\|_{\mathcal{H}_\kappa}\right)^2.$$

Then $\left|\phi_j(P_i) - \phi_j(\hat{P}_i)\right| \leq L_\phi D_2(P_i, \hat{P}_i)^{\beta_\phi}$ under (A4) and $\left\|\kappa_{P_i} - \kappa_{\hat{P}_i}\right\|_{\mathcal{H}_\kappa} \leq L_\kappa D_2(P_i, \hat{P}_i)^{\beta_\kappa}$ under (B2), hence with $\|\kappa_{P_i}\|_{\mathcal{H}_\kappa}^2 \leq 1$ under (B2), $\sum_{j=1}^{\infty} \lambda_j \leq 2a^2$ under (B3), and $\phi_j^2(P_i) \leq M_\phi^2$ under (A4),

$$\left\|\widehat{T}_i - T_i\right\|_{HS} \leq \sqrt{2}a(L_\phi + L_\kappa M_\phi)(D_2(P_i, \hat{P}_i)^{\beta_\phi} + D_2(P_i, \hat{P}_i)^{\beta_\kappa}).$$

And hence

$$\left\|\widehat{\mathscr{L}}_N - \mathscr{L}_N\right\|_{HS} \leq \sqrt{2}a(L_\phi + L_\kappa M_\phi)\frac{1}{N}\sum_{i=1}^{N}(D_2(P_i, \hat{P}_i)^{\beta_\phi} + D_2(P_i, \hat{P}_i)^{\beta_\kappa}).$$

Then under Assumption (B4), applying Lemma A.11 gives that, with probability $1 - 2\delta$,

$$\left\|\widehat{\mathscr{L}}_N - \mathscr{L}_N\right\|_{HS} \leq 2\sqrt{2}a(L_\phi + L_\kappa M_\phi)\rho(m)^{\beta'}\left(1 + \frac{2^{\max\{\beta_\phi, \beta_\kappa\}/2}\log(2/\delta)}{\sqrt{N}}\right), \tag{27}$$

i.e., $\left\|\widehat{\mathscr{L}}_N - \mathscr{L}_N\right\|_{HS}$ is bounded by a constant times $a_m^{1/2}$, after enlarging constants.

Combining the two terms and squaring gives that, with probability $1 - 3\delta$,

$$\left\|\mathscr{L} - \tilde{\mathscr{L}}_N\right\|_{HS} \leq \left\|\mathscr{L} - \hat{\mathscr{L}}_N\right\|_{HS} + \left\|\widehat{\mathscr{L}}_N - \mathscr{L}_N\right\|_{HS}$$

$$\leq \frac{4\sqrt{2}aM_\phi\log(2/\delta)}{\sqrt{N}} + 2\sqrt{2}a(L_\phi + L_\kappa M_\phi)a_m\left(1 + \frac{2^{\max\{\beta_\phi, \beta_\kappa\}/2}\log(2/\delta)}{\sqrt{N}}\right),$$

which proves the claim. $\qquad\square$

The next lemma is the same Davis–Kahan eigenspace perturbation bound as Lemma A.5, applied to the distribution-valued covariate setting.

**Lemma A.13.** *Under Assumption (B5),*

$$\|\widehat{\Pi}_s^{\mathcal{H}} - \Pi_s^{\mathcal{H}}\|_{\mathrm{op}} \leq \frac{2\|\widehat{\mathscr{L}}_N - \mathscr{L}\|_{\mathrm{op}}}{\Delta_s} \leq \frac{2\|\widehat{\mathscr{L}}_N - \mathscr{L}\|_{HS}}{r_s}.$$

*Proof.* This is Lemma A.5 applied with $X \equiv P$ and $\widehat{X} \equiv \hat{P}$. The last inequality uses $\|\cdot\|_{\mathrm{op}} \leq \|\cdot\|_{HS}$ and $r_s \leq \Delta_s$. $\qquad\square$

**Lemma A.14** (Eigenspace contribution to excess risk). *Assume Assumptions (A4) and (B2)–(B6). Let $g_s^{\mathcal{H}} = \Pi_s^{\mathcal{H}}g$ and $\hat{g}_s^{\mathcal{H}} = \widehat{\Pi}_s^{\mathcal{H}}g$. Then*

$$\|g_s^{\mathcal{H}} - \hat{g}_s^{\mathcal{H}}\|_{L_2(\mathbb{P}_P)}^2 = \mathcal{O}_P\left\{\frac{1}{r_s^2}\left(\frac{1}{N} + a_m\right)\right\}.$$

*Proof.* Using $\|h\|_{L_2(\mathbb{P}_P)}^2 \leq \lambda_1\|h\|_{\mathcal{H}_\kappa}^2$ and $\lambda_1 \leq a^2$ under Assumption (B3),

$$\left\|\hat{g}_s^{\mathcal{H}} - g_s^{\mathcal{H}}\right\|_{L_2(\mathbb{P}_X)}^2 \leq a^2\left\|\hat{g}_s^{\mathcal{H}} - g_s^{\mathcal{H}}\right\|_{\mathcal{H}_\kappa}^2.$$

Since

$$\hat{g}_s^{\mathcal{H}} - g_s^{\mathcal{H}} = (\widehat{\Pi}_s^{\mathcal{H}} - \Pi_s^{\mathcal{H}})g,$$

we have $\left\|\hat{g}_s^{\mathcal{H}} - g_s^{\mathcal{H}}\right\|_{\mathcal{H}_\kappa}^2$ being upper bounded as

$$\left\|\hat{g}_s^{\mathcal{H}} - g_s^{\mathcal{H}}\right\|_{\mathcal{H}_\kappa}^2 = \left\|(\hat{\Pi}_s^{\mathcal{H}} - \Pi_s^{\mathcal{H}})(g)\right\|_{\mathcal{H}_\kappa}^2$$
$$\leq \left\|\hat{\Pi}_s^{\mathcal{H}} - \Pi_s^{\mathcal{H}}\right\|_{op}^2 \|g\|_{\mathcal{H}_\kappa}^2$$

And then applying Lemma A.13 (with $r_s$ as a lower bound on $\Delta_s$) gives that

$$\left\|\hat{g}_s^{\mathcal{H}} - g_s^{\mathcal{H}}\right\|_{\mathcal{H}_\kappa}^2 \leq \frac{4R^2 \left\|\widehat{\mathscr{L}}_N - \mathscr{L}\right\|_{HS}^2}{r_s^2}.$$

Then we apply Lemma A.12 to get

$$\left\|\hat{g}_s^{\mathcal{H}} - g_s^{\mathcal{H}}\right\|_{\mathcal{H}_\kappa}^2 = \mathcal{O}_P\left(r_s^{-2}(N^{-1} + a_m)\right).$$

And then with $\|g\|_{\mathcal{H}_\kappa}^2 \leq R^2$ from (B6), $\left\|\hat{g}_s^{\mathcal{H}} - g_s^{\mathcal{H}}\right\|_{\mathcal{H}_\kappa}^2$ is upper bounded as

$$\left\|\hat{g}_s^{\mathcal{H}} - g_s^{\mathcal{H}}\right\|_{\mathcal{H}_\kappa}^2 = \mathcal{O}_P\left(r_s^{-2}(N^{-1} + a_m)\right).$$

and $\left\|\hat{g}_s^{\mathcal{H}} - g_s^{\mathcal{H}}\right\|_{L_2(\mathbb{P}_X)}^2$ is correspondingly bounded as

$$\left\|\hat{g}_s^{\mathcal{H}} - g_s^{\mathcal{H}}\right\|_{L_2(\mathbb{P}_X)}^2 = \mathcal{O}_P\left(r_s^{-2}(N^{-1} + a_m)\right).$$

$\square$

**Lemma A.15** (Proxy evaluation transfer). *Assume Assumptions (B2) and (B4). Let*

$$\mathcal{G}_B = \{h \in \mathcal{H}_\kappa : \|h\|_{\mathcal{H}_\kappa} \leq B\}.$$

*Then, uniformly over $h \in \mathcal{G}_B$,*

$$\mathbb{E}\left[(h(\widehat{P}) - h(P))^2\right] \lesssim B^2 a_m,$$

*and*

$$\left|\mathbb{E}\left[(Y - h(\widehat{P}))^2\right] - \mathbb{E}\left[(Y - h(P))^2\right]\right| \lesssim (1 + B^2)a_m.$$

*Proof.* For the first claim, for $h \in \mathcal{G}_B$, the reproducing property and Assumption (B2) give

$$|h(\hat{P}) - h(P)| = \left|\langle h, \kappa(\cdot, \hat{P})\rangle - \langle h, \kappa(\cdot, P)\rangle\right|$$
$$= \left|\langle h, \kappa(\cdot, \hat{P}) - \kappa(\cdot, P)\rangle\right|$$
$$\leq \|h\|_{\mathcal{H}_\kappa} \left\|\kappa_{\hat{P}} - \kappa_P\right\|_{\mathcal{H}_\kappa}$$
$$\leq B \left\|\kappa_{\hat{P}} - \kappa_P\right\|_{\mathcal{H}_\kappa}$$
$$\leq BL_\kappa D_2(\hat{P}, P)^{\beta_\kappa},$$

where the first equality is from the reproducing property, the first inequality comes from Cauchy-Schwarz, and the last inequality is due to Assumption (B2). Notice that $BL_\kappa$ is a multiplication of predefined constants. Hence this with further bounding $\mathbb{E}\left[D_2(\hat{P}, P)^{2\beta_\kappa}\right] \leq \rho(m)^{2\beta_k}$ under (B4), finally we obtain

$$\mathbb{E}\left[\left(h(\hat{P}) - h(P)\right)\right] \leq B^2 L_\kappa^2 \mathbb{E}\left[D_2(\hat{P}, P)^{2\beta_\kappa}\right]$$
$$\leq B^2 L_\kappa^2 \rho(m)^{2\beta_k} \tag{28}$$
$$\leq B^2 L_\kappa^2 a_m,$$

where the last inequality comes from the definition of $a_m$.

The second follows from the identity

$$(Y - h(\widehat{P}))^2 - (Y - h(P))^2 = \{h(P) - h(\widehat{P})\}\{2Y - h(\widehat{P}) - h(P)\},$$

Cauchy–Schwarz, boundedness of $h$, and the sub-Gaussian moment bound for $Y$, after enlarging constants in $a_m$. □

**Lemma A.16** (Estimation in the learned feature space). *Condition on $\widehat{P}_{1:N}$ and fix s. Let $\widehat{\mathcal{F}}_s = \mathrm{span}\{\widehat{\phi}_1, \ldots, \widehat{\phi}_s\}$, and let $\widehat{g}_{s,\xi}$ be the estimator in* (16). *Suppose the finite-dimensional stability event*

$$\mathcal{E}_{s,n,m} = \left\{ \|\widehat{g}_{s,\xi}\|_{\mathcal{H}_\kappa}^2 \vee \|\widehat{\Pi}_s^{\mathcal{H}} g\|_{\mathcal{H}_\kappa}^2 \le B_s^2 \right\}$$

*holds, where $B_s^2 \lesssim (r_s^2)^{-1}$. Then, on $\mathcal{E}_{s,n,m}$,*

$$\mathbb{E}\left[ (\widehat{g}_{s,\xi}(P) - g(P))^2 \mid \widehat{P}_{1:N} \right] \lesssim \inf_{\substack{h \in \widehat{\mathcal{F}}_s \\ \|h\|_{\mathcal{H}_\kappa} \le B_s}} \mathbb{E}\left[ (h(P) - g(P))^2 \mid \widehat{P}_{1:N} \right] + \frac{s}{n} + \frac{\xi}{\xi + \lambda_s^2} + \frac{1}{r_s^2} a_m.$$

*Proof.* Conditional on $\widehat{P}_{1:N}$, the features are fixed and satisfy $\|\widehat{\Phi}_s(\widehat{P})\|_2 \le 1$ by Lemma A.9. Thus the second stage is an $s$-dimensional ridge regression problem with bounded design. On $\mathcal{E}_{s,n,m}$, the same finite-dimensional oracle inequality used in Lemma A.8 gives

$$\mathbb{E}\left[ (Y - \widehat{g}_{s,\xi}(\widehat{P}))^2 \mid \widehat{P}_{1:N} \right] \le \inf_{\substack{h \in \widehat{\mathcal{F}}_s \\ \|h\|_{\mathcal{H}_\kappa} \le B_s}} \mathbb{E}\left[ (Y - h(\widehat{P}))^2 \mid \widehat{P}_{1:N} \right] + \mathcal{O}_P\left( \frac{s}{n} + \frac{\xi}{\xi + \lambda_s^2} \right).$$

By Lemma A.15, both sides can be transferred from $\widehat{P}$ to $P$ at cost

$$(1 + B_s^2) a_m \lesssim \frac{1}{r_s^2} a_m.$$

Finally, since $g$ is the $L_2(\mathbb{P}_P)$-projection of $f$ onto $\mathcal{H}_\kappa$, for every $h \in \mathcal{H}_\kappa$,

$$\mathbb{E}\left[ (Y - h(P))^2 \right] - \mathbb{E}\left[ (Y - g(P))^2 \right] = \mathbb{E}\left[ (h(P) - g(P))^2 \right].$$

Combining the preceding displays proves the lemma. □

### A.2.2. PROOF OF PROPOSITION 4.1

*Proof of Proposition 4.1.* We work on the intersection of the eigengap event in Assumption (B5) and the finite-dimensional stability event $\mathcal{E}_{s,n,m}$ in Lemma A.16; the resulting bound is an $\mathcal{O}_P$ statement. By Lemma A.16,

$$\mathbb{E}\left[ (\widehat{g}(P) - g(P))^2 \right] \lesssim \inf_{\substack{h \in \widehat{\mathcal{F}}_s \\ \|h\|_{\mathcal{H}_\kappa} \le B_s}} \mathbb{E}\left[ (h(P) - g(P))^2 \right] + \frac{s}{n} + \frac{\xi}{\xi + \lambda_s^2} + \frac{1}{r_s^2} a_m.$$

Take $h = \widehat{g}_s^{\mathcal{H}} = \widehat{\Pi}_s^{\mathcal{H}} g$. On $\mathcal{E}_{s,n,m}$, this function belongs to the feasible class in the infimum. Then, using Lemmas A.10 and A.14,

$$\|\widehat{g}_s^{\mathcal{H}} - g\|_{L_2(\mathbb{P}_P)}^2 \lesssim s^{-q} + \frac{1}{r_s^2}\left( \frac{1}{N} + a_m \right).$$

Combining the preceding displays proves (18). □

A.2.3. PROOF OF THEOREM 4.2

*Proof of Theorem 4.2.* The regression bound follows from the triangle inequality in $L_2(\mathbb{P}_P)$, exactly as in Theorem 3.3.

For the prediction bound, work on the same stability event as Proposition 4.1. Lemma A.15 applied to $h = \widehat{g}$ gives

$$\left| \mathbb{E}\left[ (Y - \widehat{g}(\widehat{P}))^2 \right] - \mathbb{E}\left[ (Y - \widehat{g}(P))^2 \right] \right| \lesssim \frac{1}{r_s^2} a_m,$$

which is absorbed into $R^2_{s,n,N,m}$. Since $Y = f(P) + \mu$ and $\mathbb{E}[\mu \mid P] = 0$,

$$\mathbb{E}\left[ (Y - \widehat{g}(P))^2 \right] = \mathbb{E}\left[ (f(P) - \widehat{g}(P))^2 \right] + \tau_0^2.$$

Applying the regression bound proves (20). □

A.2.4. PROOF OF COROLLARY 4.3

*Proof of Corollary 4.3.* Let $s_n \asymp n^{1/(q+1)}$. By the two negligibility conditions in Corollary 4.3, the operator-proxy term and the ridge shrinkage term in (18) are both $o(s_n^{-q} + s_n/n)$. Therefore Proposition 4.1 gives

$$R^2_{s_n,n,N,m} = \mathcal{O}_P\left( s_n^{-q} + \frac{s_n}{n} \right).$$

Since $s_n^{-q} \asymp n^{-q/(q+1)}$ and $s_n/n \asymp n^{-q/(q+1)}$, the result follows. □

# B. Experiment Details

In our notation, $N$ is a total number of examples and $n$ is the number of labeled examples (so we have $N - n$ unlabeled examples in our data). For all simulations, we assume we have $m$ samples for each distribution feature. In other words, our observed data would be represented as the following form.

$$\mathcal{D}^{obs} = \{(\mathcal{X}_1, y_1), \cdots, (\mathcal{X}_n, y_n), \mathcal{X}_{n+1}, \cdots, \mathcal{X}_N\},$$
$$\text{where } \mathcal{X}_i = \{X_{i1}, \cdots., X_{im}\} \text{ for } i = 1, \cdots, N.$$

## B.1. Summary of baseline group methods

We use four groups of baseline methods; three different settings for supervised learning and the other one for semi-supervised learning. The first group is supervised counterpart (**SL1**) that is able to take distribution features without any unlabeled examples; we use Support Distribution Machines (SDM) regression (Sutherland et al., 2012). The second supervised group (**SL2**) uses the first four moments and their second order interactions as features instead of taking distribution features directly; we employ LASSO$_1$, Random Forest (RF), Support Vector Machine regression (SVR), and Kernel Ridge Regression (KRR). For the third group of our supervised counterpart (**SL3**), algorithms were implemented upon all the individual samples ($n \times m$) treating each of them as independent example; we use LASSO$_2$, Convolutional Neural Network (CNN) and RF. Finally, for our semi-supervised learning counterpart (**SSL**) without treating features as a distribution, we employ Laplacian Regularized Least Squares (LapRLS) (Belkin et al., 2006) and Embed CNN (Weston et al., 2008) (E-CNN) which are known to yield the most state-of-the-art performance of semi-supervised learning. For this group, we model the case of both moment based features (as in group 2) and $N \times m$ individual samples (as in group 3), then only present the one with better out-of-sample performance.

Descriptions of these baseline group methods are also summarized in Table 4. We only include baseline methods that are known to work well in each setting.

## B.2. Parameter Selection Process

In this section we describe the parameter selection process for each method we use for the real-data simulation. Throughout the simulation we used zero regularization penalty ($\xi = 0$) to avoid excessive computational cost with understanding that the regularization might be of help to reduce overfitting and thus to obtain more stable result.

*Table 4.* Summary of baseline group methods

| Group | Methods | Distributional Features | Unlabeled Examples |
|---|---|---|---|
| Supervised Counterpart I | SDM | Yes | No |
| Supervised Counterpart II | $\text{LASSO}_1$, RF, SVR, KRR | No; Use first four moments and their interactions | No |
| Supervised Counterpart III | $\text{LASSO}_2$, CNN, RF | No; Use all the individual samples | No |
| Semi-supervised Counterpart | LapRLS, Embed CNN | No; Either moments or individual samples | Yes |
| SSDR | **Algorithm 1** | Yes | Yes |

- **SSDR**(Algorithm 1): For density estimation, we use RBF kernel with bandwidth chosen from cross-validation. For our distribution kernel, we simply use Euclidean kernel ($\kappa(f, g) = \int f(x)g(x)dx$). To determine the value of $s$, one needs to specify value of $a$ and $q$ in (A3) first, which can be estimated from empirical eigenvalue distribution. In our experiment, we found that in most cases, it is sufficient to use the value of $a \approx 1$ and $q$ slightly larger than 2 to upper bound the $\lambda_k$ for all $k$. Then we select $s$ if marginal error improvement becomes less than 0.01 within range $10^1 \leq s \leq 10^3$.

- **SDM**: We borrow code from Sutherland `https://github.com/dougalsutherland/sdm` with default setting for parameters.

- **LASSO**: We use R-package `glmnet` and built-in cross-validation procedure in the package for parameter selection.

- **SVR**: We use `sklearn.svm` in Python and use grid search to determine value of $C$ from $1 \leq C \leq 20$.

- **RF**: We use R-package `range` and built-in cross-validation procedure in the package for parameter selection.

- **KRR**: We use `sklearn.kernel_ridge` in Python with Gaussian kernel and use grid search for parameter selection.

- **CNN**: We use tensorflow library via Python to implement plain 2-layer CNN of $\{10, 10\}$ where only the first layer is convolutional with stride of 2. ADAM optimizer was used for training.

- **LapRLS**: We slightly modified the original Matlab code from Belkin and Sindhwani [1]. We use heat kernel weights $W_{ij} = e^{\|x_i - x_j\|_2^2/t}$ as our weighted edge for some suitable choice of $t$. By simple grid search, we set $\gamma_A l = 0.005$, $\frac{\gamma_I l}{(u+l)^2} = 0.05$ for the first simulation and $\gamma_A l = \frac{\gamma_I l}{(u+l)^2} = 0.005$ for the second.

- **Embed CNN**: We added a semi-supervised loss ($l_2$-norm regularizer) to the supervised loss on the entire network???s output based on the 2-layer CNN used as supervised counterpart with the same weighted edge matrix $W$.

### B.3. Synthetic Experiment: Noisy Euclidean Covariates

We also include a synthetic experiment for the noisy Euclidean covariate setting in Section 3. This experiment is designed to isolate the effect of unlabeled proxy covariates, separately from the distribution-regression structure studied in Appendix B.4. We generate a two-dimensional latent variable

$$U_i = (U_{i1}, U_{i2}) \sim \text{Unif}([-1, 1]^2),$$

and embed it into a ten-dimensional smooth nonlinear covariate

$$X_i = \big(U_{i1}, U_{i2}, \sin(\pi U_{i1}), \cos(\pi U_{i2}), U_{i1}U_{i2}, U_{i1}^2, U_{i2}^2, \sin\{\pi(U_{i1} + U_{i2})\}, \cos\{\pi(U_{i1} - U_{i2})\}, U_{i1}^3 - U_{i2}^3\big)^\top \in \mathbb{R}^{10}.$$

---

[1] `http://manifold.cs.uchicago.edu/manifold\_regularization/software.html`

*Table 5.* Prediction Error for different choices of $n$ and $N$ for the synthetic dataset of beta distribution

| $n$ | $N$ | | | | | | |
|---|---|---|---|---|---|---|---|
| | 0 | 50 | 100 | 150 | 200 | 250 | 300 |
| 50 | 0.3010 | 0.2780 | 0.2766 | 0.2259 | 0.1560 | 0.1668 | 0.1501 |
| 60 | 0.2466 | 0.1897 | 0.1543 | 0.1357 | 0.0959 | 0.0860 | 0.0942 |
| 70 | 0.2051 | 0.1362 | 0.0552 | 0.0546 | 0.0390 | 0.0461 | 0.0444 |
| 80 | 0.1551 | 0.0962 | 0.0612 | 0.0359 | 0.0346 | 0.0290 | 0.0301 |
| 90 | 0.1432 | 0.0733 | 0.0526 | 0.0244 | 0.0251 | 0.0197 | 0.0153 |

The response is generated from the smooth regression function

$$Y_i = f(X_i) + \epsilon_i, \qquad f(X_i) = \sin(\pi U_{i1}) + U_{i2}^2 + \frac{1}{2}U_{i1}U_{i2}, \qquad \epsilon_i \sim N(0, 0.1^2).$$

The learner does not observe $X_i$. Instead, it observes the noisy proxy

$$\widehat{X}_i = X_i + \tau W_i, \qquad W_i \sim N(0, I_{10}),$$

with $\tau \in \{0.10, 0.40\}$, representing mild and stronger proxy noise.

We fix the number of labeled examples at $n = 50$, vary the total number of proxy covariates $N \in \{100, 300, 1000\}$, and evaluate prediction on an independent test sample. We compare three methods: (i) SSDR, which learns spectral features from all $N$ proxy covariates and fits the second-stage regression using only the labeled observations; (ii) a labeled-only spectral baseline, which uses the same spectral ridge pipeline but learns the features only from the $n$ labeled proxy covariates; and (iii) an oracle latent baseline, which applies the same procedure using the unobserved latent covariates $X$. Performance is measured by normalized RMSE on the test set, and the reported standard deviations are computed across repeated Monte Carlo replications.

The results are reported in Table 2. When proxy noise is mild, increasing $N$ substantially improves SSDR and moves it closer to the oracle latent baseline. When proxy noise is stronger, the improvement from increasing $N$ is smaller and begins to saturate, consistent with the proxy perturbation term in Theorem 3.2.

## B.4. Synthetic Experiment: Skewness of Beta distribution

To check the validity of our algorithm, we use the Beta distribution in which the first parameter is chosen randomly from certain range of numbers whereas the second parameter is fixed. Specifically, we have $\beta eta(a, 3)$ where $a$ is chosen randomly from *Uniform*[3, 20]. Then sample of size $m = 30$ is drawn from $\beta eta(a, 3)$ for each $a$. This process is repeated 300 times. The first $N$ sample sets constitute the training set and consequently the remaining sets constitute the testing set.

In our case $b$ remains fixed at 3 hence $f$ is a simply function of $a$ which is a distributional feature. In this example we use the kernel $\kappa(f, g) = \int f(x)g(x)dx$, and calculate the error by using the root mean squared error (RMSE) of the target function at the predicted values of the parameter for the test samples. The results are presented in Table 5. The results show that in general the prediction error decrease in $N$ and $n$ as expected and indeed the algorithm achieves very small prediction errors even for moderate values of $(N, n, m)$.

## B.5. Mass Measurements of Galaxy Clusters

In this real-data application, we estimate mass of galaxy clusters using distribution of galaxy velocity. All the cluster catalog are gleaned from a publicly available halo catalog of the Multidark Simulation[2]. We randomly choose 80% of the data for training, 10% for validation whenever necessary for parameter selection, and use the rest of 10% for testing.

Clusters are the most massive gravitationally-bound systems in the Universe. Since a cluster mass is not directly observable, once clusters are identified mass measurements are needed to map observable cluster properties to the underlying mass. (Ntampaka et al., 2015) studied the modern machine learning approach for cluster dynamical mass measurements. The power law says Halo mass $M$ can be related to galaxy line-of-sight (LOS) velocity ($v_{los}$) dispersions. Unlike earlier studies that only use summary statistics of mass measurements, they take advantage of using the information encoded in the probability distribution of $v_{los}$ based on the method from (Sutherland et al., 2012). It is known that measuring the galaxy velocity is

---

[2]https://www.cosmosim.org/

much easier than estimating the galaxy mass, hence it is reasonable to assume we have more unlabeled data in hand. Our algorithm can be well applied to estimate the cluster mass from this distribution variable.

Since the unit of galaxy mass is very huge, we calculate the prediction error as scaled root-mean-squared-error. We randomly select $n$ out of $N = 1632$ as the number of labeled examples, and vary the ratio of $n$ to $N$. Here, $m_i$ corresponds to the number of galaxies in the $i^{th}$ cluster which is fairly large. In Table 1, we note that the proposed algorithm (**SSDR**) generally outperforms other baseline methods. We recognize the prediction error is quite big in this simulation since we only use one distributional variable in our model.

To compute eigenvalues efficiently, we use Lanczos Method. If the algorithm did not converge (which happened only once in this galaxy dataset), we additionally imposed sparsity structure on the kernel matrix and implemented relevant algorithms (e.g. a function found in `scipy.sparse`).

### B.6. Default Risk and Stock Returns

A firm defaults when it fails to service its debt obligations. Therefore, default risk induces lenders to require from borrowers a spread over the risk-free rate of interest. This spread is an increasing function of the probability of default of the individual firm. Thus, the relationship between default risk and stock returns has important implications for the risk-reward trade-off in financial markets. Various attempts have been made in financial literature to assess the effect of default risk on equity returns or vice versa.

However, a relationship between stock returns and default risk is still controversial. For example, (Garlappi & Yan, 2011; Garlappi et al., 2008) support the negative relationship between default risk and stock returns, whereas (Chava & Purnanandam, 2010) argue the opposite. Even though many of studies recognize the importance of realized stock return distributions on future default risk, most of them merely rely on running multiple cross-sectional regressions with simple cumulative return or average return for analysis.

For the simulation we collect stocks' daily returns up to month $T$ with 90-day windows and aim to predict the default risk of time $T + 1$ (or $T + 2$ in case that the majority of data are not available), and this prediction is rolling over entire time interval (5 years). To compute company's default risk, we collect data from FnGuide [3]. Each vendor adopts slightly different model to estimate company's 1-year default risk basically from CDS spread. In fact, there are significant missing values for default risk at each time point, which motivates us to use the proposed algorithm.

We use same set of baseline methods as the galaxy cluster dataset. In fact, even though the screening standard is top 500 in company's market value there are quite substantial portion of missing values each time point (roughly 18% of entire data on average). So composition of labeled samples can be slightly varying from time to time but we do not think this largely affect our analysis. Moreover, many papers in finance literature treat stock returns as *i.i.d.*, so the use of kernel density estimation to estimate $\hat{p}$ is also justified in this analysis. Since in general the default risk is very small, we use scaled RMSE to measure the prediction error. Again, we confirm our method generally outperforms the other baseline methods.

---

[3]Dataguide by FnGuide: http://www.dataguide.co.kr/DG5web/eng/index.asp

