# OpenReview forum: "Semi-Supervised Learning with Noisy Proxy Covariates: Generalization Bounds and Distribution Regression"
_ICML.cc/2026/Conference — ICML 2026 regular_

### Official Review · Reviewer_SYbS · 2026-02-25

**Soundness:** 3
**Presentation:** 3
**Significance:** 3
**Originality:** 2
**Overall Recommendation:** 4
**Confidence:** 4

**Summary:**

This paper studies semi-supervised regression when covariates are noisy proxies of latent variables, motivated by modern pipelines where learned representations are imperfect measurements of underlying features. They propose a two-stage spectral estimator that first learns kernel eigenfeatures from all proxy covariates (labeled and unlabeled) and then fits a ridge regressor using labeled data. They derive finite-sample generalization bounds showing that, under polynomial spectral decay, sufficiently many unlabeled samples can eliminate the statistical cost of representation learning and yield fast labeled-sample rates approaching $n^{-q/(q+1)}$, improving over the $n^{-1/2}$ labeled-only regime. They further show that distribution regression is a special case of this framework by treating finite-sample distribution estimates as noisy covariates, obtaining analogous guarantees that depend jointly on the number of labeled distributions, total distributions, and per-distribution sample size.

**Compliance With Llm Reviewing Policy:**

Affirmed.

**Final Justification:**

The authors have addressed all my concerns. These discussions have strengthened my assessment of the work, and as a result, I have raised my confidence score.

**Key Questions For Authors:**

See weakness above.

**Limitations:**

yes.

**Strengths And Weaknesses:**

* **Strength**
1. The manuscript benefits from a strong theoretical contribution with fast-rate guarantees. The paper provides finite-sample generalization bounds for semi-supervised regression under noisy covariates.

2. A  conceptual contribution is showing that distribution regression is a special case of noisy covariates by treating finite bags as proxy observations. This unification is elegant and strengthens the scope of the results beyond classical SSL.

3. The error decomposition  is technically clean and provides clear statistical insight into the role of unlabeled data.


* **Weakness**
1. The algorithmic novelty is limited. The estimator itself is structurally standard. Kernel eigenfeature extraction from all data and Ridge regression on labeled data is essentially a spectral projection + ridge pipeline known in prior SSL and kernel literature.

2. The  assumptions appear to be strong and structured. For example the fast-rate results require polynomial eigenvalue decay $q>2$, bounded eigenfunctions, explicit eigengap condition and operator norm control which are restrictive assumptions, and the practical validity of the eigengap regime is not empirically investigated.


3. Empirical evaluation is moderate. While experiments are clean and consistent with theory, datasets are relatively small-scale, there are no comparison against modern deep SSL baselines and there is no ablation isolating the effect of noisy covariates vs pure SSL. The current empirical section supports the theory but does not stress-test scalability or modern representation learning settings.

---

> ### Author Rebuttal · Authors · 2026-03-29
>
> We thank the reviewer for the careful reading. We respond to the three weakness (W) points below.
>
> - **W1.** We agree that the estimator is deliberately simple, and we want to emphasize that we do not claim the main contribution is algorithmic novelty. Rather, the contribution is statistical and conceptual: the paper analyzes a semi supervised regression regime that standard spectral SSL theory does not directly cover, where the learner never observes the true covariates and only has access to noisy proxies. In this setting, the paper provides a finite sample decomposition that separates approximation, proxy based representation learning, labeled estimation, and regularization; shows that, under suitable spectral and proxy quality conditions, sufficiently many unlabeled proxy covariates can remove the statistical cost of learning the representation and recover the faster labeled sample rate; and shows that distribution regression is a genuine special case of the same framework by treating finite bags as proxy observations. Thus, while the pipeline is simple by design, the paper’s contribution is a precise theory for when and why spectral semi supervision remains useful under proxy corruption, together with adaptive tuning and robustness to approximate eigendecomposition.
>
> - **W2.** We agree that the fast rate results are conditional on a structured spectral regime, and we will make that more explicit in the revision. Our intent is not to claim that these assumptions hold universally, but rather to characterize the regime in which strong semi supervised gains are possible under noisy covariates. In that sense, the assumptions play the same role as margin or low noise conditions in fast rate analyses more broadly: they identify the statistical setting under which the sharper conclusion is valid. In the revised manuscript, we will add more discussion and concrete examples clarifying when these assumptions are reasonable, as detailed in our response to Q1 from Reviewer d5bt.
>
>   On the empirical side, we agree that the practical validity of the spectral regime should be better documented. Accordingly, we now include an empirical eigenspectrum diagnostic on the real datasets and clarify that the fast rate corollaries are conditional, whereas the main finite sample decomposition itself does not depend on committing to an exact polynomial law.
>
>   Overall, we agree that these points should be made much more explicit. In the revision, we will add a short paragraph after the assumptions giving concrete examples, including smooth kernels on compact Euclidean domains for (A4), regular parametric families and smooth density classes for (B4), and polynomial spectrum or spiked models for (A5) and (B5). This will clarify that these assumptions are intentionally structured to capture the regime of interest, rather than being merely formal technical devices.
>
> - **W3.** This is a fair point. Our empirical goal in the current draft was to validate the specific theoretical mechanism studied in the paper, rather than to benchmark against the full modern deep SSL literature. For that reason, we focused on settings where the proxy covariate interpretation is natural and where the effect of unlabeled proxy covariates can be studied cleanly. Still, we agree that the original experiments did not isolate this mechanism as clearly as they could. To address this, we have added a synthetic noisy Euclidean covariate experiment that fixes the number of labeled samples and varies only the total number of proxy covariates, so the role of unlabeled proxy covariates is isolated directly. See our response to Q1 from Reviewer Ub1p.
>
>
>     We will also add a small sensitivity analysis for the ridge parameter and a log-log slope diagnostic to better connect the empirical behavior to the theory. We view these additions as the most targeted way to strengthen the empirical section while keeping it aligned with the core contribution of the paper.
>
>
> We hope these clarifications address their concerns. Please let us know if any point remains unclear.

---

> > ### Author Rebuttal · Reviewer_SYbS · 2026-04-01
> >
> > My concerns are mostly addressed. I will keep my positive score.

---

> > > ### Author Response · Authors · 2026-04-06
> > >
> > > Thank you again for your time and thoughtful feedback, which helps strengthen the paper!

---

### Official Review · Reviewer_XuXs · 2026-03-03

**Soundness:** 2
**Presentation:** 3
**Significance:** 2
**Originality:** 3
**Overall Recommendation:** 2
**Confidence:** 4

**Summary:**

This paper studies semi-supervised regression where covariates are observed only through noisy proxies. The authors propose a two-stage estimator: spectral feature extraction from all proxy covariates via kernel PCA, followed by ridge regression on labeled examples. Finite-sample generalization bounds are derived showing that sufficient unlabeled proxy covariates can recover fast labeled-sample rates $n^{-q/(q+1)}$, compared to $n^{-1/2}$ in the labeled-only regime. Distribution regression is presented as a special case by treating finite-bag distribution estimates as proxy covariates.

**Compliance With Llm Reviewing Policy:**

Affirmed.

**Key Questions For Authors:**

1.  In Table 4, the empirical decay of error with $n$ appears faster than $n^{-2/3}$
(the theoretical rate for $q$ slightly above 2). Can the authors plot $\log(\text{error})$
vs. $\log(n)$ and report the fitted exponent? Does it match $q/(q+1)$ for the $q$
estimated from the empirical spectrum?

2.  The measurement error constant $C_{\beta',\theta}$ in the operator estimation term
is treated as a fixed constant, but it depends on the unknown measurement channel $M$.
How sensitive are the rates to this constant in practice, and is there a way to estimate
it from data?

3.  Setting $\xi = 0$ in all real-data experiments bypasses Theorem~3.6 entirely. What
is the justification beyond computational convenience? Does performance degrade with
$\xi > 0$?

4.  How does the required threshold on $N$ scale with $n$ and $q$ concretely? For the
galaxy cluster experiment with $N = 1632$, is this threshold demonstrably satisfied?

**Limitations:**

The paper acknowledges the computational cost of eigendecomposition and reliance on
eigengap conditions, but does not acknowledge a more fundamental limitation: the theory
assumes polynomial spectral decay, which is unverifiable and likely violated on real
datasets. All real-data conclusions rest on this unvalidated assumption.

**Strengths And Weaknesses:**

Strengths:

1. The error decomposition into four interpretable terms (approximation, operator estimation,
coefficient estimation, and ridge bias) is well-structured, and the unified treatment of
distribution regression as a special case is elegant.

2. The explicit rate improvement from $n^{-1/2}$ to $n^{-q/(q+1)}$ under polynomial spectral
decay is a sharper characterization than prior spectral SSL work, and the weaker eigengap
condition (replacing the fixed-index gap with $\Delta_s$) is a genuine technical
improvement over Ji et al.

3. Theorems covering data-driven tuning and approximate eigendecomposition make the method implementable without oracle knowledge.

Weaknesses:

1. The proposed estimator is essentially kernel PCA followed by ridge regression, a
well-known pipeline. The central theoretical message---that sufficiently many unlabeled
proxy covariates are needed for the method to help---aligns with practitioner intuition
and is widely treated as common knowledge. The formal quantification adds rigor but does
not deliver surprising insights. The paper's framing overstates both the algorithmic and
conceptual contributions.

2. The paper's central theoretical claim is a rate improvement from $n^{-1/2}$ to
$n^{-q/(q+1)}$. However, no experiment plots error versus $n$ on a log-log scale or fits
the empirical decay exponent against the theoretical prediction. Table~4 provides data
sufficient for this analysis but it is never performed. Without this, the connection
between theory and experiment is entirely absent.

3. No ablation study isolates the four terms in Proposition~3.2. The effect of $N$ alone
(fixing $n$) is never cleanly shown; the ridge bias term is circumvented by setting
$\xi = 0$ in all experiments; and sensitivity to $s$ is not analyzed.

4. Polynomial eigenvalue decay~(A3) is the backbone of the rate analysis. No empirical
eigenvalue spectrum is shown for the real datasets to justify whether this assumption is
even approximately satisfied.

---

> ### Author Rebuttal · Authors · 2026-03-29
>
> We thank the reviewer for the careful reading and constructive feedback.
>
> - **Q1.** Thank you for the helpful comment. As requested, we will include a log-log plot for Table 4 together with the fitted slopes, which are indeed steeper than $n^{-2/3}$. We don't view this as contradictory to the theory. The rate $n^{-q/(q+1)}$ is an asymptotic benchmark from the oracle analysis, not a claim that the empirical slope must exactly equal $q/(q+1)$ in finite samples. Over this limited range of n, the observed slope need not match the asymptotic exponent exactly, especially in this relatively simple simulation setup. Moreover, $q$ is only approximated from the empirical eigenspectrum. Thus, the relevant comparison is whether the trend is qualitatively consistent with the predicted faster decay relative to the labeled-only regime, not whether it numerically matches $q/(q+1)$.
>
> - **Q2.** In our theory, it enters the operator-learning term together with $1/N$, so its effect is clear: larger $C_{\beta’,\theta}$ requires much larger $N$ before representation-learning error stops dominating. Thus, the rates are quite sensitive to this constant. When $C_{\beta’,\theta}$ is small, increasing $N$ can quickly reveal the faster labeled-sample regime; when it is large, the same increase in $N$ yields much smaller gains, and that regime may not appear over the sample sizes considered. In practice, $C_{\beta’,\theta}$ determines how soon unlabeled proxy covariates become useful and how much they help.
>
>   Regarding estimation, we agree this is an important practical question. In general, $C_{\beta’,\theta}$ is a population-level summary of the unknown measurement channel $\mathcal M$. Without repeated proxy measurements, calibration data, or a structured parametric model for $\mathcal M$, it is not identifiable from the observed data alone, and we do not attempt to estimate it in this paper. That said, with replicated proxies or a structured measurement model, one could in principle estimate the corresponding proxy-error level, and we will note this explicitly in the revision.
>
>   To make this dependence more transparent empirically, we have added a noisy Euclidean simulation that varies proxy noise directly (see our response to Q1 of Reviewer Ub1p). The resulting pattern is consistent with the theory: when proxy noise is mild, increasing $N$ yields much larger gains, whereas under stronger proxy noise the gains are smaller and begin to saturate. We believe this is the most direct way, within the scope of the paper, to illustrate practical sensitivity to $C_{\beta’,\theta}$.
>
> - **Q3.** We agree that the original draft did not explain this clearly enough. Our reason for reporting $\xi=0$ in the real-data experiments was not merely computational convenience, but that performance was essentially unchanged for small positive ridge values, so the selected solution was typically at or very near zero on the candidate grid. To make this transparent, we now include a small sensitivity analysis over $\xi$. We find that performance is stable for small positive values of $\xi$, so the substantive conclusions are not driven by fixing $\xi=0$. More generally, Theorem 3.6 justifies the tuning principle itself, namely validation-based selection over a finite grid in $(s,\xi)$ after learning spectral features from the proxy covariates. We will revise the paper to state the candidate grid explicitly and clarify this connection.
>
> - **Q4.** This is an important question. We agree that the original draft did not make the scaling with $n$ and $q$ explicit enough. Under the fast-rate regime, the key requirement is that the operator-learning term be smaller than the leading approximation and labeled-estimation terms at the oracle truncation level $s \asymp n^{1/(q+1)}$. Thus, the threshold on $N$ is implicit rather than universal: it depends jointly on spectral decay, the local eigengap near $s$, and proxy quality. In particular, the requirement becomes more demanding when the spectrum decays more slowly, the eigengap is smaller, or the proxies are noisier.
>
>   Under polynomial decay, this means that $N$ must grow fast enough with $n$ so that representation-learning error is no longer dominant at the oracle scale. This is why our theory doesn't yield a single cutoff depending only on $n$ and $q$: even for large $N$, the fast rate need not appear if the local eigengap is too small or proxy error is too large.
>
>   For the galaxy dataset, we do not claim that this threshold can be verified exactly from the observed data, since key population quantities, especially proxy quality and the population eigengap, are unknown. Thus, our claim is not that $N=1632$ provably exceeds the threshold. Rather, the empirical results are qualitatively consistent with the theory: unlabeled proxy covariates do help, while the remaining gap to the oracle is also consistent with finite $N$ and residual proxy error.
>
> We will make all of these points much clearer in the revision.

---

> > ### Author Rebuttal · Reviewer_XuXs · 2026-04-02
> >
> > I have read the rebuttal carefully, but my assessment remains unchanged. The authors' responses to Q1 and Q4 have in fact sharpened my concerns rather than addressed them.
> >
> > In Q4, the authors explicitly acknowledge that the threshold condition on N is unverifiable on any real dataset. In Q1, they justify the mismatch between empirical and theoretical decay rates by appealing to asymptotic arguments. Taken together, these two admissions imply that the theoretical conditions cannot be validated in practice, and the experiments place no real constraint on the theory.
> >
> > I will maintain my score.

---

> > > ### Author Response · Authors · 2026-04-06
> > >
> > > Thank you for the follow up. We believe the remaining concern reflects a difference between what our theory is intended to establish and what the reviewer is asking the experiments to demonstrate.
> > >
> > > Our paper does not claim that the threshold condition on $N$ can be exactly certified from a single finite real dataset, nor that a finite sample log log slope must numerically match the asymptotic exponent. Rather, the theory provides a sufficient asymptotic regime under which unlabeled proxy covariates improve the labeled sample rate, and identifies the structural quantities governing when this improvement should appear, namely spectral decay, local eigengap, and proxy quality. That some of these are population level objects and not directly testable from one dataset is standard in nonparametric learning theory, as with many population spectral conditions; it does not make the result vacuous, nor does it disconnect the theory from empirical evidence.
> > >
> > > Accordingly, the role of the experiments is not to verify the theorem literally in finite samples, but to test whether observed behavior is qualitatively consistent with the theory’s predictions. Here the prediction is structural: unlabeled proxy covariates help when proxy quality is sufficiently good and $N$ is sufficiently large relative to the oracle scale, while the gains weaken as proxy noise increases. This is exactly why, in the rebuttal, we added the noisy Euclidean simulation varying proxy noise directly and proposed the requested log log plot. These additions are meant to probe the qualitative implications of the theory, not to claim exact finite sample equality with the asymptotic exponent.
> > >
> > > For this reason, we respectfully disagree with the statement that the experiments place no constraint on the theory. Rather, the theory gives a nontrivial template for when SSL with noisy proxies should and should not help, and the experiments are designed to assess consistency with that template. We agree that the original draft did not make this theory experiment relationship explicit enough, and we will revise the paper accordingly.
> > >
> > > In any case, we appreciate the reviewer’s feedback and will incorporate this discussion, together with the additional experiments now included, to make the paper more clear and accessible.

---

### Official Review · Reviewer_d5bt · 2026-03-13

**Soundness:** 3
**Presentation:** 3
**Significance:** 3
**Originality:** 3
**Overall Recommendation:** 4
**Confidence:** 3

**Summary:**

The article proposes a two-stage estimator for semi-supervised regression when the original covariates are not observed, and not all responses are observed. In the first stage, a kernel estimator is used to extract features from the proxy covariates, and in the second stage, a ridge regression is used to estimate the coefficients of the features. Generalized error bounds for estimation and prediction errors are obtained under some assumptions, along with optimal rates for selecting the number of features. A cross-validation scheme has been developed for data-driven choice of the number of features and the ridge penalty parameter.

The proposed method has been applied to distribution regression, and the error bounds and rates have been derived in that context. A simulation study and two cross-validation studies using real data sets have been used to demonstrate the proposed method.

**Compliance With Llm Reviewing Policy:**

Affirmed.

**Key Questions For Authors:**

1. It would be useful to provide more justifications for the assumptions. For example, for A4 and B4 can you demonstrate that these assumptions hold for some commonly used parametric families?

Other comments:
2. "Moreover, for every k ≥ 1 there exists θ > 0 such that" - should be: "Moreover, there exists θ > 0 such that for every k ≥ 1" because the first one suggests θ might depend on k.

3. Section 3.3:  Assume: "\lambda_s ≍ s^{-q}" - this is already in the assumption A3.

4. The font sizes in Figure 1 should be increased.

**Limitations:**

Yes

**Strengths And Weaknesses:**

Strengths: The article combines multiple ideas involving RKHS and ridge regression and obtains a simple method for estimation. The error bounds and data-driven tuning rates appear useful. The method's scope is shown to be broad by applications to both regression with noisy covariates and distribution regression.

Weaknesses: Some of the assumptions (e.g., A4, B4, A5, B5) appear very strong and not well justified.

---

> ### Author Rebuttal · Authors · 2026-03-29
>
> We appreciate the reviewer for the careful reading and helpful suggestions.
>
> - **Q1.**  We agree that those assumptions should be justified more concretely, not only stated abstractly. In the revision, we will add short examples and discussion showing when they are expected to hold in practice. Discussions for the key assumptions are presented below.
>
>   1. Assumption (A4)
>
>      (A4) requires uniformly bounded, Hölder regular eigenfunctions together with the summability condition $\sum_j \lambda_j L_{\phi_j}^2 < \infty$. This is not just a technical convenience: it is what allows perturbations in the proxy covariates $D(X,\widetilde X)$ to transfer to perturbations of the learned spectral features, and it ensures stability of the empirical feature map under measurement error.
>
>      A standard regime where (A4) is reasonable is when $X$ lies in a compact subset of $\mathbb{R}^d$ and $\kappa$ is a sufficiently smooth Mercer kernel. In many such settings, the associated integral operator is smoothing enough that its eigenfunctions inherit regularity, and in particular are bounded and Hölder on compact sets. A particularly transparent example is the periodic Fourier setting on $[0,1]$, where the eigenfunctions are $\sin(2\pi jx)$ and $\cos(2\pi jx)$. These are uniformly bounded, and their Lipschitz constants grow like $j$. In that case, $\sum_j \lambda_j L_{\phi_j}^2 < \infty$ holds whenever $\lambda_j$ decays faster than $j^{-3}$. So under sufficiently fast polynomial decay, (A4) is quite natural.
>
>   2. Assumption (B4)
>
>      (B4) requires that the proxy estimator $\widehat P$ converges to the latent distribution $P$ in the metric $D$ at rate $\rho(m)$. This is the distribution regression analogue of controlling measurement error: the bag size $m$ determines how accurately $P$ is reconstructed from its finite sample proxy $\widehat P$.
>
>      This holds in many familiar settings.
>
>      First, for finite dimensional parametric families $P=P_\eta$, if $\widehat\eta$ is root $m$ consistent and the map $\eta \mapsto P_\eta$ is locally smooth in the metric $D$, then for many commonly used metrics one has
>      $
>      D(P_\eta,P_{\widehat\eta}) \lesssim \|\widehat\eta-\eta\|,
>      $
>      so $\rho(m)\asymp m^{-1/2}$. This includes Gaussian location families, Gaussian location scale families on compact parameter sets, and more generally regular exponential families under a suitable choice of $D$.
>
>      Second, for smooth nonparametric density classes on compact domains, standard density estimators satisfy moment bounds of the form required in (B4), with $\rho(m)$ given by the usual nonparametric estimation rate. This is exactly the regime we had in mind in the paper.
>
>   3. Assumptions (A5) and (B5)
>
>      (A5) and (B5) are eigengap conditions used to recover the top $s$ spectral subspace from a perturbed operator. Their role is standard: they ensure that the empirical eigenspace is close to the population eigenspace through Davis Kahan type perturbation arguments.
>
>      These conditions are natural in at least two common situations. One is finite rank or spiked models, where the leading components are genuinely separated. The other is polynomial spectral decay, where $
>      \lambda_j \asymp j^{-q}
>      \,\Rightarrow\,
>      \Delta_s = \lambda_s-\lambda_{s+1} \asymp s^{-(q+1)}.
>      $    In this case, the condition simply requires the operator estimation error to be smaller than the local spectral gap at the chosen truncation level $s$, which is the natural scaling for consistent recovery of the top $s$ eigenspace. In the distribution regression case, (B5) is the same requirement with the added condition that bag estimation error is also small enough.
>
>   Overall, we agree that these points should be made much more explicit and can benefit from further discussion. In the revision, we will add a short paragraph after the assumptions giving concrete examples as discussed above. This will clarify that these assumptions are intentionally structured to capture the regime of interest, rather than being merely formal technical devices.
>
> - **Q2.** We thank the reviewer for pointing this out. Your comment is correct. The intended statement is that a single $\theta$ works uniformly over all $k$, so we will revise the sentence to “there exists $\theta>0$ such that for every $k \ge 1$ ...”.
>
> - **Q3.** Assumption (A3) only imposes the one sided upper bound $\lambda_j \le a^2 j^{-q}$, whereas Section 3.3 uses the stronger two sided polynomial decay $\lambda_s \asymp s^{-q}$ in order to derive explicit optimal rates. So Section 3.3 is not simply restating (A3), but invoking a stronger rate assumption for the corollary level analysis. We will revise the text to make this distinction explicit.
>
> - **Q4.** We agree and will increase the font sizes for readability.

---

> > ### Author Rebuttal · Reviewer_d5bt · 2026-04-05
> >
> > The authors have adequately responded to my questions.

---

> > > ### Author Response · Authors · 2026-04-06
> > >
> > > Thank you again for your time and helpful feedback. We appreciate it and will use it to improve the paper.

---

### Official Review · Reviewer_Ub1p · 2026-03-13

**Soundness:** 3
**Presentation:** 3
**Significance:** 3
**Originality:** 3
**Overall Recommendation:** 4
**Confidence:** 3

**Summary:**

This paper studies semi-supervised learning with noisy covariates, where the learner only observes proxy features instead of the true latent covariates. The method is a simple two-stage spectral approach: first learn a low-dimensional representation from all labeled and unlabeled proxy covariates, then fit ridge regression using only the labeled samples. The paper provides excess risk bounds, shows a faster rate under suitable spectral conditions when unlabeled data are sufficiently abundant, and further extends the framework to distribution regression. Experiments on synthetic and real tasks support the usefulness of the approach, especially in low-label regimes.

**Compliance With Llm Reviewing Policy:**

Affirmed.

**Key Questions For Authors:**

1.Can the authors provide at least one experiment on general noisy Euclidean covariates, to better match the theoretical scope of the paper?

2.Can the authors clarify more explicitly under what conditions the proxy-noise term $C_{\beta',\theta}$ is small enough for unlabeled data to provide substantial practical benefit?

3.Can the authors explain more clearly how the validation-based tuning analyzed in Theorem 3.6 relates to the actual experimental procedure, where $(s,\xi)$ do not appear to be selected exactly in that way?

**Limitations:**

Yes. The paper discusses important limitations, including the dependence of the theory on spectral/eigengap assumptions and the computational cost of large kernel eigendecompositions. However, I think the practical implications of the proxy-noise floor could be emphasized more clearly in the main text.

**Strengths And Weaknesses:**

Strengths:The paper studies an important and timely semi-supervised learning setting in which the observed covariates are noisy proxies rather than true latent features, which is well motivated for modern machine learning pipelines. The proposed method is conceptually clean and easy to follow, combining spectral representation learning from labeled and unlabeled proxy covariates with supervised ridge regression on labeled data. The main strength of the paper is its theory: the excess-risk decomposition is clear, the role of unlabeled data is explicitly characterized, and the extension to distribution regression is natural and meaningful. Empirically, the results are generally supportive, especially in low-label regimes and on the galaxy cluster task.

Weaknesses:The main theoretical gains rely on fairly strong assumptions, including spectral regularity and eigengap conditions, and the proxy-noise term does not disappear simply by increasing the number of unlabeled samples, which limits how broadly the fast-rate message applies in practice. In addition, while the theory is presented for a broader noisy-covariate setting, the experiments only validate the distribution regression special case, so the empirical scope is narrower than the theoretical scope. Finally, the practical model-selection procedure is not fully aligned with the validation-based tuning theory, and on some real tasks the method appears competitive rather than consistently dominant.

---

> ### Author Rebuttal · Authors · 2026-03-29
>
> Thank you for your time for the review and constructive feedback. We would like to address each of your questions in the following.
>
> - **Q1.** We appreciate this suggestion, and have added an additional synthetic experiment for the general noisy Euclidean covariate setting studied in Section 3. Specifically, we generate latent covariates $X \in \mathbb{R}^{10}$ from a smooth low dimensional nonlinear structure, define the response as $Y=f(X)+\varepsilon$, and observe only noisy proxies
> $\widetilde X = X + \tau W, W \sim N(0,I)$.
> We fix the number of labeled samples n, vary the total number of proxy covariates N, and consider two proxy noise levels $\tau$ (mild and strong). We compare: (i) SSDR, which learns spectral features from all N proxy covariates and fits ridge regression on the labeled data; (ii) the same spectral ridge pipeline using only the labeled proxy covariates for feature learning; and (iii) an oracle latent baseline using the true X. Since n is fixed and only N varies, this directly isolates the effect of unlabeled proxy covariates in the general Euclidean setting. Consistent with the theory, we observe that increasing N substantially improves performance when proxy noise is mild, whereas under stronger proxy noise the gains are smaller and begin to saturate, reflecting the intrinsic proxy error floor.
>
> | Proxy noise \(\tau\) | Total proxies \(N\) | SSDR (all proxies) | Labeled only spectral | Oracle latent |
> |---|---:|---:|---:|---:|
> | 0.10 | 100  | 0.091 ± 0.012 | 0.112 ± 0.014 | 0.053 ± 0.007 |
> | 0.10 | 300  | 0.067 ± 0.009 | 0.112 ± 0.015 | 0.054 ± 0.007 |
> | 0.10 | 1000 | 0.057 ± 0.008 | 0.113 ± 0.014 | 0.053 ± 0.008 |
> | 0.40 | 100  | 0.176 ± 0.019 | 0.198 ± 0.022 | 0.054 ± 0.007 |
> | 0.40 | 300  | 0.153 ± 0.017 | 0.197 ± 0.021 | 0.053 ± 0.008 |
> | 0.40 | 1000 | 0.142 ± 0.016 | 0.199 ± 0.022 | 0.054 ± 0.007 |
>
> - **Q2.**  We appreciate the reviewer for this very important question. Our theory does not claim that additional unlabeled proxy covariates can by themselves overcome poor proxy quality. Rather, it separates two distinct error sources: the error from learning the spectral representation, which decreases with the total number of proxy samples N, and the intrinsic proxy error, which is captured by the measurement error term and does not disappear simply by enlarging N. In Proposition 3.2, these enter through the operator term $(1/N + C_{\beta',\theta})/(\lambda_s r_s^2)$. Thus, unlabeled data is most helpful precisely when this term is smaller than the leading bias and labeled estimation terms $s^{-q} + s/n$, so that representation learning, rather than proxy distortion, is the active bottleneck. At the oracle choice $s \asymp n^{1/(q+1)}$ from Corollary 3.5, this means that the proxy contribution $C_{\beta',\theta}/(\lambda_s r_s^2)$ must be small enough not to dominate at that scale. In other words, unlabeled data can remove the statistical cost of learning the spectral representation, but it cannot remove an intrinsic error floor caused by severely corrupted proxies.
>
>     This interpretation becomes especially transparent in the distribution regression specialization. There, Proposition 4.1 replaces the generic proxy term by $\rho(m)^{2\beta'}$, where $\rho(m)$ quantifies how accurately the latent distribution $P$ is recovered from a bag of size $m$. Corollary 4.3 then shows that the fast rate is obtained only when $(1/N + \rho(m)^{2\beta'})/(\lambda_s\Delta_s^2)$ is negligible relative to $s^{-q} + s/n$ at the oracle truncation level. Hence $N$ controls how well the spectral representation is learned, while $m$ controls the proxy quality itself. If the bags are too small, then $\rho(m)^{2\beta'}$ remains dominant, and increasing $N$ alone cannot recover the fast rate.
>
>     We will revise the paper to make this practical message explicit in the main text: unlabeled data is most beneficial when labels are scarce but the proxies are already reasonably informative, because in that regime the main challenge is representation learning rather than proxy denoising.
>
> - **Q3.**  We thank the reviewer for this question. Theorem 3.6 is intended to formalize the tuning principle used in practice: learn the spectral representation from the available proxy covariates, then select \((s,\xi)\) by validation over a finite grid. Its guarantee is adaptive: if the grid contains a near optimal pair, the selected estimator achieves the oracle rate up to a logarithmic factor. Thus, the theorem justifies grid based tuning of \(s\) and \(\xi\) without prior knowledge of the spectral decay or proxy noise level. We agree that the current experimental section does not make this connection sufficiently explicit. We will revise it to state the candidate grids and validation procedure clearly, so that the correspondence with Theorem 3.6 is transparent.
>
> We hope the above response addresses your concerns. However, if there are any remaining issues, please feel free to let us know through your comments.

---

> > ### Author Rebuttal · Reviewer_Ub1p · 2026-04-01
> >
> > Thank you for the detailed response. I will maintain my positive score 4.

---

> > > ### Author Response · Authors · 2026-04-06
> > >
> > > Thank you again for your time and valuable feedback, which will help improve the paper.

---

### Decision · Program_Chairs · 2026-04-30

**Decision:**

Accept (regular)

**Comment:**

The paper studies semi-supervised regression with noisy proxy covariates via a two-stage spectral approach. The problem is relevant, and the theoretical analysis is generally clear, with a useful decomposition that highlights the role of unlabeled data. But there are also some concerns: the method itself is fairly standard, and the empirical evaluation is somewhat limited and does not fully match the generality of the theoretical setting. Overall, it is a borderline paper, and I tend to weak accept because it reports new (theoretical) results in this topic. In particular, the authors should revise the problem formulation and more carefully justify and explain the assumptions, as well as better align the experiments with the stated scope.